

# ASIS v1.0: an adaptative solver for the simulation of atmospheric chemistry

Daniel Cariolle [1,2], Philippe Moinat [1], Hubert Teyssèdre [3,†], Luc Giraud [4], Béatrice Josse [3], and Franck Lefèvre [5]

[1]Climat, Environnement, Couplages et Incertitudes, UMR5318 CNRS/Cerfacs, Toulouse, France
[2]Météo-France, Toulouse, France
[3]Centre National de Recherches Météorologiques, UMR3589 CNRS/Météo-France, Toulouse, France
[4]Institut National de Recherche en Informatique et en Automatique, Talence, France
[5]Laboratoire Atmosphères, Milieux, Observations Spatiales, CNRS/UPMC/UVSQ, Paris, France
[†]deceased, April 2013

*Correspondence to:* D. Cariolle (daniel.cariolle@cerfacs.fr)

**Abstract.** This article reports on the development and tests of the Adaptative Semi-Implicit Scheme (ASIS) solver for the simulation of atmospheric chemistry. To solve the Ordinary Differential Equation systems associated with the time evolution of the species concentrations, ASIS adopts a one step linearized implicit scheme with specific treatments of the Jacobian of the chemical fluxes. It conserves mass and has a time stepping module to control the accuracy of the numerical solution. In idealized box model simulations ASIS gives results similar to the higher order implicit schemes derived from the Rosenbrock's and Gear's methods. When implemented in the MOCAGE CTM and the LMD Mars GCM the ASIS solver performs well and reveals weaknesses and limitations of the original semi-implicit solvers used by these two models. ASIS can be easily adapted to various chemical schemes and further developments are foreseen to increase its computational efficiency, and to include the computation of the concentrations of the species in aqueous phase in addition to gas phase chemistry.

## 1 Introduction

In Chemical Transport Models (CTMs) or General Circulation Models (GCMs) the description of atmospheric chemistry has rapidly increased in complexity. Early model developments were devoted to the study of the stratospheric and upper tropospheric compositions focussing on the gas phase reactions that control the ozone distribution. Emphasis has afterwards been put on tropospheric chemistry due to its oxidant properties and its possible impact on climate via the lifetime of several greenhouse gases and the distribution of secondary-formed aerosols.

Large scale models include now chemical schemes that deal with about hundred species and several hundreds of reactions in gas-phase and in heterogeneous phases (solid and liquid). Most of those species undergo transport processes, like advection, diffusion and convection. As a result the models include the resolution of complex coupled systems which cannot be handled in a single operator. In practice the various processes are decomposed in a series of operators that are solved numerically in sequence. For example the time evolution of the species are first calculated taking into account advection, then diffusion,





convection, and so on. Among these processes the evolution of the species due to chemical transformations is a key component of the models.

The models have to solve coupled ODE systems that describe the adopted chemical mechanism. These ODE systems are of the non-linear form:

$$\partial C / \partial t = f(t, C) = P(t, C) - L(t, C).C \tag{1}$$

where $C$ represents the vector of the local species concentrations, $P(t, C)$ and $L(t, C)$ the production and loss term matrices. The stiffness of the systems comes from the wide range of values that can take the production and loss terms. Small values of the loss term correspond to stable species having long lifetimes (e.g. $CH_4$, $N_2O$, ...) whereas large values correspond to radical species (e.g. $O(^1D)$, OH, Cl, ..) with short lifetimes. Typical atmospheric situations lead to species lifetimes ranging from milliseconds to years. Since the other physical processes can change the conditions and compositions of the air masses (i.e. surface emissions, transport at all scales, day-night transitions, etc...) the chemical system is often out of chemical equilibrium and the ODE system to be solved can be very stiff. Adequate algorithms must then be used for its resolution.

For atmospheric applications some numerical properties of those algorithms should be particularly sought for:

i) *Mass conservation*. Atmospheric models are often integrated for long term simulations, up to several decades for global climate simulations, and small trends and anomalies are investigated. Any bias or trend in the atmospheric composition due to numerical algorithms must therefore be avoided. It is therefore essential that the algorithms chosen to solve the chemical systems preserve mass. All the atoms or elementary groups of atoms (e.g. nitrogen oxides) must be conserved.

ii) *Accuracy*. It is of course always desirable to obtain a numerical solution as accurate as possible, although the uncertainties associated with the other operators and the fact that they are integrated successively in time introduces a significant degree of inaccuracy. This leads also to transient evolutions in the chemical system, especially for the short lived radicals, that have no real physical basis. It is not always necessary to obtain a very accurate numerical solution during those transient evolutions if they do not last long and have little impact on the solutions for the other longer lived species. The key point is to design an algorithm where the accuracy can be chosen a priori by the user and controlled during the course of the numerical integration.

iii) *Positivity*. It is highly desirable to maintain positivity of the concentrations. Otherwise instability might arise when coupled to other operators dealing with advection or convection of the minor species. Some algorithms maintain positivity of the solution by construction, others introduce clipping of the negative values at the expense of local mass conservation. Negative values can be tolerated if they are small and transient, and if they have little impact on the algorithms used to account for the other physical processes.

iv) *Adaptability and flexibility*. The adopted solver should cope with a variety of chemical mechanisms with the possibility to easily add or remove species and reactions. It is also desirable to let to the user a minimum of free parameters to tune. The solver should also run efficiently on a large variety of computers without having to rewrite large parts of the code. This can be obtained with extensive use of mathematical libraries that are often optimized for the computer being used.

This article describes a solver for the simulation of gas-phase atmospheric chemistry, the Adaptative Semi-Implicit Scheme (ASIS), that has most of the desirable properties discussed above. Section 2 gives the basic formulation of the scheme, section



gives results from box model simulations and comparison with other state of the art algorithms, and sections 4 and 5 detail the implementation of ASIS within the MOCAGE CTM (Michou et al., 2002; Josse at al., 2004; Teyssèdre et al., 2007) and the GCM of planet Mars (Lefèvre et al., 2004) of the Laboratoire de Météorologie Dynamique (LMD). Possible future extensions of the solver are discussed in the last section.

## 2 ASIS: description of the chemical solver

### 2.1 Implicit discretisation and numerical methods

The integration of Eq. (1) cannot be done using a simple one-step explicit scheme with the left-hand side terms evaluated at time t, as numerical stability would required to use timesteps lower than the shortest species lifetime. Since some radicals have lifetimes lower than a few milliseconds in the atmosphere too many iterations would be required to obtain simulations for hundred or more days with affordable computer time. Several explicit methods have been developed to address this issue which are based on classification of the species according to their lifetimes. For instance with the QSSA method (Hesstvedt et al., 1978) the fast species with lifetimes much lower than the timestep are often assumed to be at equilibrium, $C = P/L$, the intermediate species are obtained using an exponential solution of Eq. (1) and the long-lived species are computed using the simple explicit solution. Other explicit schemes gain in accuracy with the use of multi-step algorithms with predictor-corrector evaluations of the concentration at time $t + \delta t$. For instance the CHEMEQ solver of Young and Boris (1997) with subsequent developments by Mott et al. (2001). Limitations of these explicit schemes are that they often do not conserve mass and that the choice of species classifications is somewhat arbitrary. Mass conservation can be improved using the technique of "species lumping" where additional equations are introduced for linear combinations of species concentrations to reduce the stiffness or enforce conservation for a chemical family. The drawback of those approaches is that the algorithm becomes problem dependent and requires a very good knowledge of the chemical system especially when updating the constant rates or the list of reacting species.

One possibility to increase the timestep is to treat part of the right-hand side of Eq. (1) implicitly, for instance keeping the evaluations of P and L at time $t$ but C at time $t + \delta t$:

$$C^{t+1}(I + L(t,C)\delta t) = C^t + P(t,C)\delta t \tag{2}$$

where $C^{t+1}$ is the concentration vector of the species at time $t + \delta t$. The second term of the left hand side of this equation is diagonal so its numerical resolution is straightforward. With this discretisation the numerical solution is positive and unconditionally stable, but mass conservation is not maintained.

One way to alleviate this problem is to discretize Eq. (2) fully implicitly in time using the simple backward Euler method:

$$C^{t+1}(I + L(t+1,C)\delta t) = C^t + P(t+1,C)\delta t \tag{3}$$

Resolution of Eq. (3) requires an evaluation of the terms $L(t+1,C)$ and $P(t+1,C)$ that can be obtained using $C^{t+1}$ from the resolution of Eq. (2). In practice Eq. (3) is solved iteratively with successive evaluations of $C^{t+1}$, for instance using the





iterative Newton method. Still, the mass conservation can only be obtained if a good convergence of the solution is reached and additional constraints such as species lumping or equilibrium assumptions for the shorter lived species are often used to increase accuracy and to speed up convergence. A scheme of this type is for example used within the MOCAGE CTM.

The implicit methods described above to solve the ODE chemical system are all one timestep: only concentrations at time $t$ are used to evaluate the concentrations at time $t + \delta t$. Although the numerical stiff ODE field is largely developed and precise ODE solvers are available and have been used for atmospheric chemistry problems, many of them are multi-steps or multi-stages. Consequently, several evaluations of $C$ at various past or intermediate timesteps are used to obtain the concentration at time $t + \delta t$. A direct extension of the simple backward Euler method is to use higher order backward differentiation formula (BDF) to solve Eq.(1). Based on that approach Verwer (1994) has developed the TWOSTEP atmospheric chemical solver,

which uses a second order BDF formula combined with a Gauss-Seidel iteration technique to solve the resulting implicit system. This solver can be very efficient but it is not naturally mass conserving.

Mass conserving, multi-step or multi-stage and high order accurate implicit methods exist to solve the ODE stiff system. Among the methods based on BDF, Gear's predictor-corrector method has been adapted to atmospheric chemical systems, for exemple the SMVGEAR code (Jacobson and Turco, 1994) implemented in the GEOSCHEM CTM (Bey et al., 2001).

More recently the Rosenbrock's method (Rosenbrock, 1963), is becoming widely used in atmospheric chemistry modelling (Sandu et al., 1997) despite the fact that its computational cost is still rather high compared to approaches based on low order BDF methods. The implementation in chemical models of Rosenbrock's and other high-order methods has been eased by the development of the Kinetic PreProcessor (KPP) by Sandu et Sander (2006), which allows the choice of an integration method and generates the adequate codes accordingly.

When the chemical scheme involves more than a hundred species and over two hundreds of reactions, the implicit multi-stage methods are still computationally expensive, especially if they are to be used within global 3D models with horizontal resolutions of the order of $1° \times 1°$, with several tens of vertical levels and for simulations lasting for several years. The increase of the computational cost comes from the need to solve at each stage a linear system of the order of the number of species, and this cost varies non-linearly (often quadratically) with the number of species.

## 2.2   Formulation of the ASIS solver

The approach adopted for ASIS is to restrict the algorithm to a single implicit step combined with a specific evaluation of the Jacobian matrix of the chemical fluxes, $J = f'(C) = \partial f / \partial C$.

The starting point comes from the decomposition of chemical tendencies in three terms:

$$\frac{\partial C_k}{\partial t} = \sum_{l,m} \sigma K_{l,m} C_l C_m - D_k C_k + F_k \tag{4}$$

with $\sigma = -1$ if $m = k$ and $l \neq k$, $\sigma = -2$ if $l = m = k$, and $\sigma = 1$ if $l$ and $m \neq k$ , where $C_k$ is the concentration of the $k$ species.

The first term of the right-hand side corresponds to the chemical productions or destructions due to first order reaction rates with constants $K$. The second term arises from thermal decompositions and/or photodissociations of species with a rate $D$, and




the last term accounts for external tendencies that come from other physical processes than chemistry. For example the surface emissions affecting the lowest levels of the model will result in species tendencies $F$.

The time discretization of Eq. (4) is then performed with a semi-implicit scheme for the first term adapted to each reaction and timestep, an implicit discretization for the second one and the external tendencies are assumed to be constant over the timestep $\delta t$:

$$\frac{(C_k^{t+1} - C_k^t)}{\delta t} = \sum_{l,m} \sigma K_{l,m}[\Delta_{l,m}^t C_l^t C_m^{t+1} + (1 - \Delta_{l,m}^t)C_l^{t+1}C_m^t] - D_k C_k^{t+1} + F_k \tag{5}$$

with $0 \leq \Delta_{l,m}^t \leq 1$

Eq.(5) can be recast with terms containing species concentrations at time $t+1$ on the left-hand side and the others on the right-hand side:

$$C_k^{t+1} - \sum_{l,m} \sigma K_{l,m}\delta t[\Delta_{l,m}^t C_l^t C_m^{t+1} + (1 - \Delta_{l,m}^t)C_l^{t+1}C_m^t] + D_k \delta t C_k^{t+1} = C_k^t + F_k \delta t \tag{6}$$

that can be reformulated in a matrix form:

$$(I - M\delta t)C^{t+1} = C^t + F\delta t \tag{7}$$

with the matrix $M$, an approximation of the Jacobian $J$, containing species concentrations at time $t$ and values of $\Delta_{l,m}$ evaluated also at time $t$.

Compared to other one step semi-implicit schemes like SIS (i.e. Ramarosson et al. 1994), one specificity of our scheme lies in the evaluation of $\Delta_{l,m}^t$. Let us consider the system of a single reaction between species $C_l$ and $C_m$ with a reaction rate constant $K_{l,m}$. If the initial values of the concentrations are equal, $C_l^0 = C_m^0 = C^0$, the exact solution of the system gives a hyperbolic decay for the concentrations:

$$C_l(t) = C_m(t) = C^0/(1 + K_{l,m}C^0 t) \tag{8}$$

This solution is obtained exactly using the discretization given by Eq. (6) with $\Delta_{l,m} = 1/2$. If $C_l^0 >> C_m^0$ the evolution of the lowest concentration $C_m$ shows a quasi exponential decay with an e-folding time $\tau = 1/(K_{l,m}C_l^0)$ while the concentration $C_l$ reaches its steady state value $C_l^0 - C_m^0$ that does not depart strongly from its initial value. In that case, in order to maximize the timestep and to increase the stability of the scheme, there is advantage in treating the evolution of the shorter lived species $C_m$ as implicitly as possible by giving more weight to the term $C_l^t C_m^{t+1}$ in Eq. (6). This is obtained if $\Delta_{l,m}$ tends to 1. Those simple considerations lead us to introduce the following function for $\Delta_{l,m}$ that depends on the concentrations at time $t$:

$$\Delta_{l,m}^t = (C_l^t)^\beta / ((C_l^t)^\beta + (C_m^t)^\beta) \tag{9}$$

with $\beta \geq 1$. With this formulation the value of $\Delta_{l,m}$ has the required properties: $\Delta_{l,m} = 1/2$ if $C_l = C_m$ and $\Delta_{l,m} \to 1$ if $C_l \gg C_m$. The value of $\beta$ controls the sensiviy of $\Delta_{l,m}^t$ as a function of the concentrations. Large values of $\beta$ favor the implicit



treatment for the lowest concentrations. For the situations studied in this paper the numerical simulations did not show a large sensitivity to this parameter, which was fixed to 1 hereafter.

Furthermore, the use of Eq. (9) to calculate $\Delta_{l,m}^t$ and evaluate $M$, the approximate Jacobian matrix, gives interesting properties to our scheme:

– The oscillations from odd to even timesteps that can appear in the numerical solution of Eq. (6) when the semi-implicit scheme is centered and symmetrical (i.e. Shure and Rosset, 1994), as would be if the fixed value $\Delta_{l,m}^t = 1/2$ was adopted, is damped with the evaluation of $\Delta_{l,m}^t$ by Eq (9).

   – Since the largest terms contributing to the evolution of the shortest lived species are treated implicitly the system increases in stability. Larger timesteps can be used and positive values for the concentrations are more easily preserved.

– All the species are treated in the same manner without any a priori considerations on lifetimes or abundances. For instance in the case of the Earth composition $O_2$ is treated like the other species even if its chemical sources and sinks are negligible. Since the concentration of $O_2$ is much larger than the other species concentrations, any species reacting with $O_2$ will be treated implicitly. This is the case for example of atomic oxygen $O$ reacting with $O_2$ to form $O_3$. The corresponding term in Eq. (6) for the $O$ tendency will be $\Delta O_2^t O^{t+1} + (1 - \Delta)O_2^{t+1}O^t$ which reduces to $O_2^t O^{t+1}$ since

$\Delta = (O_2^t)^\beta / ((O_2^t)^\beta + (O^t)^\beta) \cong 1$. The option to treat all the species in the same manner simplifies the programming of the scheme and allows the solver to be easily adapted to various chemical systems. An example is given in section 5 with the simulation of the atmospheric composition of the planet Mars.

Once the matrix $M$ is evaluated and the timestep is determined (see the next section), the solver computes the solution to the system of linearized equations (7), which becomes a possible computational bottleneck. Our approach is to use standard

methods and well optimized software libraries.

Our baseline option is to use the direct solver DGESV of the Lapack library that solves system (6) by LU decomposition. Therefore no extra specific routine associated with the chemical mechanism is needed and the optimization on the computer used is left to the implementation of the Lapack library. As reported below this option works well and gives accurate results even for comprehensive mechanisms involving hundreds of species or more.

To reduce the computational cost other options for the resolution of the linear system have been investigated. Two iterative solvers have been tested. The first one is an implementation of the Gauss-Seidel algorithm. This algorithm has been used with success to solve stiff systems from chemical kinetics (Verwer et al., 1994; Menut et al., 2013). For the cases studied in the following sections the Gauss-Seidel algorithm was found to be efficient with a good rate of convergence in most cases. Although in specific situations where the system is largely driven out of equilibrium, for instance during day-night transitions

and for large surface emissions, the number of iterations could increase by one order of magnitude to obtain the required accuracy.

A second iterative algorithm has been implemented, the generalized minimal residual method (GMRES). The method approximates the solution by a vector in a Krylov subspace with minimal residual norm. The Arnoldi iteration algorithm is used





to find this vector. The GMRES method was developed by Saad and Schultz (1986) and further described by Saad et al. (2003). In order to accelerate the convergence, preconditioning techniques are used. An efficient one was obtained by introducing the matrix $B$ using the lower triangular part of the $A = I - M\delta t$ matrix to compute an approximation of $A^{-1}$ and apply GMRES to the solution of the right-preconditioned linear system $ABC^* = C^t + F\delta t$ where $C^{t+1} = BC^*$. For the implementation discussed hereafter the GMRES method needs less iterations than the Gauss-Seidel one, especially in situations where the Gauss-Seidel algorithm shows slower convergence, and was found to speed up the computation by at least a factor 2 compared to the DGESV implementation.

## 2.3 Time stepping

Since the time discretization adopted to solve system (7) is first order accurate, the choice of the timestep $\delta t$ is important to obtain a solution with a desired accuracy. In our applications the evolution of the species over rather large time intervals $\Delta t$ is required. $\Delta t$ is determined by other physical processes than chemistry, for instance advection, convection or vertical diffusion, and is often too large to be used directly to solve Eq. (7) without encountering numerical instabilities and loss of accuracy. For example in the 3D model results discussed in section 4, the time interval $\Delta t = 15$ minutes is determined by horizontal advection whereas the chemical timestep has to be decreased to a few seconds in situations where the chemical state is driven far from a quasi steady state.

Therefore a variable stepsize strategy has to be implemented with the time interval $\Delta t$ divided in $n$ successive integrations of the chemical system with timesteps $\delta t_n$.

The choice of $\delta t_n$ is done iteratively using a strategy similar to the one described by Verwer (1994). First a local error indicator $E$ is computed:

$$E^{k+1} = \max{}_m(|\frac{2}{(\gamma+1)}(\gamma C_m^{k+1} - (1+\gamma)C_m^n + C_m^{n-1})|/W_m) \tag{10}$$

with

$$W_m = ATOL + RTOL.C_m^n \tag{11}$$

where $\gamma = \delta t^{k+1}/\delta t_n$, $\delta t^{k+1}$ is a first guess timestep, $C_m^{k+1}$ the concentration of the species $m$ at the iteration $k+1$, $ATOL$ and $RTOL$ are absolute and relative error tolerance. $C_m^{k+1}$ is evaluated using Eq. (2) with $C_m^n$ as initial concentration and the timestep $\delta t^{k+1}$.

$E^{k+1}$ depends on the curvature of the solution, a measure of the departure of the solution from linearity. If $E^{k+1} \leq 1$ the timestep $\delta t^{k+1}$ is adopted ($\delta t_{n+1} = \delta t^{k+1}$) otherwise a new timestep is estimated by:

$$\delta t^{k+2} = \max(0.1, \min(2.0, 0.8/\sqrt{E^{k+1}}))\delta t^{k+1} \tag{12}$$

Then a new value $C_m^{k+2}$ is evaluated followed by the computation of $E^{k+2}$, and so on until convergence. In practice the convergence is obtained within a few iterations, less than 5 in the cases reported thereafter. Those iterations have a low computational





cost because the resolution of Eq.(2) at each iteration involves only diagonal matrices. Once the value of $\delta t_{n+1}$ is determined the concentration $C_m^{n+1}$ is obtained by resolution of Eq.(7).

For the first iteration species concentrations at two consecutive times and a first guess timestep are needed. To avoid storing concentrations at consecutive times we assume that at the beginning of the iterative process the system is in a steady state, $C_m^n = C_m^{n-1}$ in Eq. (10), and the first guess timestep is set to its largest possible value $\Delta t$. To secure the iterative process a minimum timestep, $\delta t_{min}$, is also prescribed in order to limit the number of iterations. The value of this minimum timestep if left to the user who has to choose a value consistent with the error tolerance parameters.

## 3   Tests and validation

To validate and evaluate the performances of ASIS and the associated numerical codes several case studies have been used. All these cases are based on the RACMOBUS chemical scheme used within the MOCAGE CTM. RACMOBUS is a combination of the REPROBUS scheme adapted to the stratosphere and the free troposphere (Lefèvre et al., 1994), and the RACM scheme (Stockwell et al.,1997) that treats the urban polluted earth atmosphere with addition of volatile organic compounds, VOCs, and their degradation products. Table 1 lists the chemical species taken into account, the overall scheme includes about 120 species linked by 200 gas-phase reactions and photodissociations. The photodissociation rates are calculated every 15 minutes using the Tropospheric Ultraviolet and Visible (TUV) radiation model version 5.2 (Madronich and Flocke, 1998) for conditions corresponding to the equinox at $30°$ latitude.

Two test cases are used to evaluate the accuracy and performance of the ASIS scheme. The first one is based on the FLUX test case described by Crassier et al. (2000). It corresponds to a ground level situation in a urban polluted area. The list of species and fluxes emitted at the surface is given in table 2. The emissions are injected in a boundary layer with a 2000 m constant thickness weighted by an emission factor of 0.6. This leads to a constant tendency $F$ in Eq. (4) for the emitted species. The initial concentrations are given in table 3, the atmospheric temperature is set to 298 K and the ground pressure is 1000 hPa.

The second case, STRATO, is representative of situations encountered in the middle stratosphere. The initial concentrations for this case are given in table 3. The atmospheric temperature is 215 K and the pressure is 50 hPa. For both cases the integration starts at midnight, stops 24 h after, and the photodissociation rates are updated every 15 minutes.

### 3.1   The FLUX case

To assess the performances of ASIS two reference simulations have been obtained for the FLUX case using Rosenbrock's and Gear's BDF solvers (referred hereafter as R1 and G1). Those solvers use respectively the ode23s and ode15s codes from the Matlab ODE suite (Shampine and Reichelt, 1997, Ashino et al., 2000). For the Rosenbrock's scheme a 3 stage algorithm is used and the simulations are third order accurate. For the Gear's scheme the third order accurate option was also chosen. The relative tolerance RTOL was set to 0.001 and the absolute tolerance ATOL equals $10^4$ molecules $cm^{-3}$ for all species. The same FLUX case is integrated using the ASIS solver. In this simulation, noted A1, ASIS uses a RTOL value of 0.001 and a



**Table 1.** List of species used for the box model simulations. The upper part of the table lists the species active in the free troposphere and the stratosphere. The lower part lists additional VOC species or generic species involved in the RACM mechanism (Stockwell et al.,1997).

| |
|---|
| $O(^1D), O(^3P), O_2, O_3,$ |
| $N, N_2O, NO, NO_2, NO_3, N_2O_5, HNO_2, HNO_4, HNO_3(gas\&solid),$ |
| $CH_4, CH_2O, CH_3, CH_4O, CH_3O, CHO, CH_4O_2, CH_3O_2, CO, CO_2$ |
| $H_2, H_2O(gas\&solid), H, OH, HO_2, H_2O_2,$ |
| $SO_2, H_2SO_4, DMS, SULFATE$ |
| $CCl_4, CFC-(11\&12\&113\&114\&115), HCFC-22,$ |
| $HA-(1202\&1211\&1301), CH_3Cl, CHCl_3, CH_3CCl_3,$ |
| $Cl, Cl_2, ClO, OClO, ClO_2, Cl_2O_2, HOCl, HCl, ClONO_2$ |
| $CH_3Br, CHBr_3,$ |
| $Br, Br_2, BrO, HBr, HOBr, BrONO_2, BrCl$ |
| ACO3, ADDC, ADDT, ADDX, ALD, API, APIP, |
| CLS, CSLP, DCB, DIEN, ETE, ETEP, ETH, ETHP, |
| GLY, HC3, HC3P, HC5, HC5P, HC8, HC8P, HKET, ISO, ISOP |
| KET, KETP, LIM, LIMP, MACR, MGLY, MO2 |
| OLI, OLIP, OLND, OLNN, OLT, OLTP, ONIT, OP1, OP2 |
| PAA, PAN, PHO, TCO3, TOL, TOLP, TPAN, |
| UDD, XO2, XYL, XYLP |

minimum timestep of 1s. For the resolution of the linear system associated with ASIS, the DGESV code of the lapack library is used. A second simulation A2 has been obtained with the same settings as A1 but with a higher relative tolerance value of 0.01, and a third one A3 with a tolerance value of 0.025. For all experiments the FLUX case is integrated over 24 h.

Figures 1 and 2 show the evolution of some key species for each experiment and the relative differences from the R1 experiment. Those results are representative of all the species. As expected the R1, G1 and A1 simulations give very close results. R1 and G1 show relative differences below 0.1 % consistent with the value chosen for RTOL. A1 results are comparable with differences in the 0.1-0.2 % range, except at the beginning of the simulation when the chemical state is out of equilibrium and during day-night transitions. In those situations the differences between A1 and R1 or G1 can reach 0.5 %. As expected from the choice of a higher value for RTOL, the A2 experiment shows less accuracy but is still in the range of 0.5% compared to the other experiments. The A3 experiment have differences below 2% with the other experiments. For most of the atmospheric simulations an accuracy below 1% is sufficient for the longer lived species, and even larger values are acceptable for short lived species if they are transient, given the uncertainties in the representation of the other processes and the inaccuracies introduced by their resolution by a series of successive operators.

The efficiency of ASIS can be evaluated by comparison of the mean timesteps (table 4). For simulations R1 and G1 the mean timesteps are between 25 and 40 s. Since ASIS uses a first order scheme to maintain good accuracy the mean timestep is lowered, in the 5 s range for the A1 experiment. However ASIS is a one stage scheme (only one linear system is solved by





**Table 2.** VOC emissions in the FLUX test case

| Species | Emission |
| --- | --- |
| | ($10^{10}$ molecules $cm^{-2}s^{-1}$) |
| NO | 121.29 |
| CO | 2500 |
| $CH_4$ | 802 |
| ETH | 6.25 |
| HC3 | 37.67 |
| HC5 | 44.43 |
| HC8 | 19.14 |
| ETE | 22.33 |
| OLT | 39.67 |
| OLI | 6.37 |
| TOL | 9.02 |
| $CH_2O$ | 5.77 |
| ALD | 14.45 |
| KET | 5.70 |
| XYL | 14.55 |
| CSL | 3.68 |

**Table 3.** Initial conditions for the FLUX and STRATO test cases

| Species | STRATO | FLUX |
| --- | --- | --- |
| | vmr | vmr |
| $O_3$ | $1.0 \ 10^{-6}$ | $50 \ 10^{-9}$ |
| $CO_2$ | $330 \ 10^{-6}$ | $330 \ 10^{-6}$ |
| $N_2O$ | $300 \ 10^{-9}$ | $310 \ 10^{-9}$ |
| NO | $1.0 \ 10^{-9}$ | $2.0 \ 10^{-9}$ |
| $NO_2$ | $0.3 \ 10^{-6}$ | $1.0 \ 10^{-9}$ |
| $HNO_3$ | $4.0 \ 10^{-9}$ | $0.5 \ 10^{-9}$ |
| $CH_4$ | $1.4 \ 10^{-6}$ | $1.6 \ 10^{-6}$ |
| CO | $20 \ 10^{-9}$ | $150 \ 10^{-9}$ |
| HCl | $2.5 \ 10^{-9}$ | $1.0 \ 10^{-12}$ |
| $ClONO_2$ | $0.3 \ 10^{-9}$ | $- - -$ |
| BrO | $15 \ 10^{-12}$ | $1.0 \ 10^{-13}$ |

timestep) compared to R1 and G1 that need 3 or more stages. The amount of computation is therefore comparable. When the

relative tolerance is increased the mean timestep of ASIS increases. For the A2 experiment it is 25 s, identical to R1, and up





**Table 4.** Mean timesteps for the FLUX test case

| R1 | G1 | A1 | A2 | A3 |
|----|----|----|----|----|
| $25s$ | $39s$ | $4.7s$ | $25s$ | $49s$ |

to 49s for A3. Since for most of atmospheric simulations a relative tolerance of 0.01 to 0.025 seems to be sufficient the ASIS solver gives acceptable solutions with less computation than the higher order schemes.

For the A1 and A2 experiments ASIS uses the DGESV code for the resolution of the linear systems. To save computational time two iterative solvers have been tested, one using the Gauss-Seidel algorithm, the other the GMRES method. Both solvers used the same criterion for convergence (tolerance for convergence set to $10^{-14}$). For the GMRES method the preconditioning technique described in section 2.1 is implemented. With those settings the experiment A2 has been repeated.

The results are practically identical to the solution obtained using the DGESV code. Figure 3 shows for example differences below 0.02 % for the simulation of $O_3$. The simulation with the Gauss-Seidel algorithm shows good efficiency in terms of mean number of iterations, but requires 6 to 10 times more iterations when the system is driven out of equilibrium during day-night transitions. In the present simulations, using GMRES was found more stable and efficient with less than 10 iterations needed to solve the linear systems and twice less computational time compared to the simulation using DGESV.

From the simulations of this FLUX case, which is rather representative of situations encountered in polluted earth boundary layers, it can be concluded that the ASIS solver performs well compared to higher order schemes when moderate accuracy is required. Apart from tolerance parameters and the choice of a minimum timestep no specific tuning is required. The one step implicit scheme gains in efficiency when coupled to the GMRES iterative solver used for the resolution of the linear systems.

## 3.2 The STRATO case

The STRATO case differs from the FLUX case in the dominant chemical regimes involved. In the FLUX case the VOC decomposition during day and night dominates the system. With the STRATO case the chemistry is dominated by NOx, HOx, Clx catalytic cycles and the ozone content. The stiffness of the system is less stringent and rapid variations in the concentrations of the species are tightly linked to the variations of the insulation at sunrise and sunset.

For this case two simulations have been performed. The first one, R2, uses the Rosenbrock's algorithm with settings similar to experiment R1. For the second one, A4, the ASIS solver is used with settings similar to experiment A2 and with the iterative linear solver GMRES. The two simulations show results consistent with the findings for the FLUX case. The mean timestep are very similar for both experiments, 49 s for R2 and $41.4$ s for A4. As expected the timestep decreases in ASIS at sunrise and sunset when the stiffness of the system is at maximum. Figure 4 shows the number of timesteps for every 15 minutes interval for the 24 hour simulation of experiment A4. Apart from the very beginning of the simulation that starts in a situation out of equilibrium, the largest values are found at sunrise and sunset in the 200 range. It corresponds to timesteps of about $4.5s$.

In terms of accuracy, the A4 experiment gives results that depart less than $1\%$ compared to R2. This is consistent with the chosen value of 0.01 for RTOL. Figure 5 shows that the lowest accuracy is found at sunrise and sunset when the short





**Figure 1.** FLUX case. Time evolution of selected species ($O_3$, $NO_2$, $NO_3$, OH) for the R1, G1, A1 and A2 experiments. The left column shows the mixing ratios and the right one the differences in % relative to the R1 experiment. The color code is the following: blue for G1, orange for A1, red for A2.





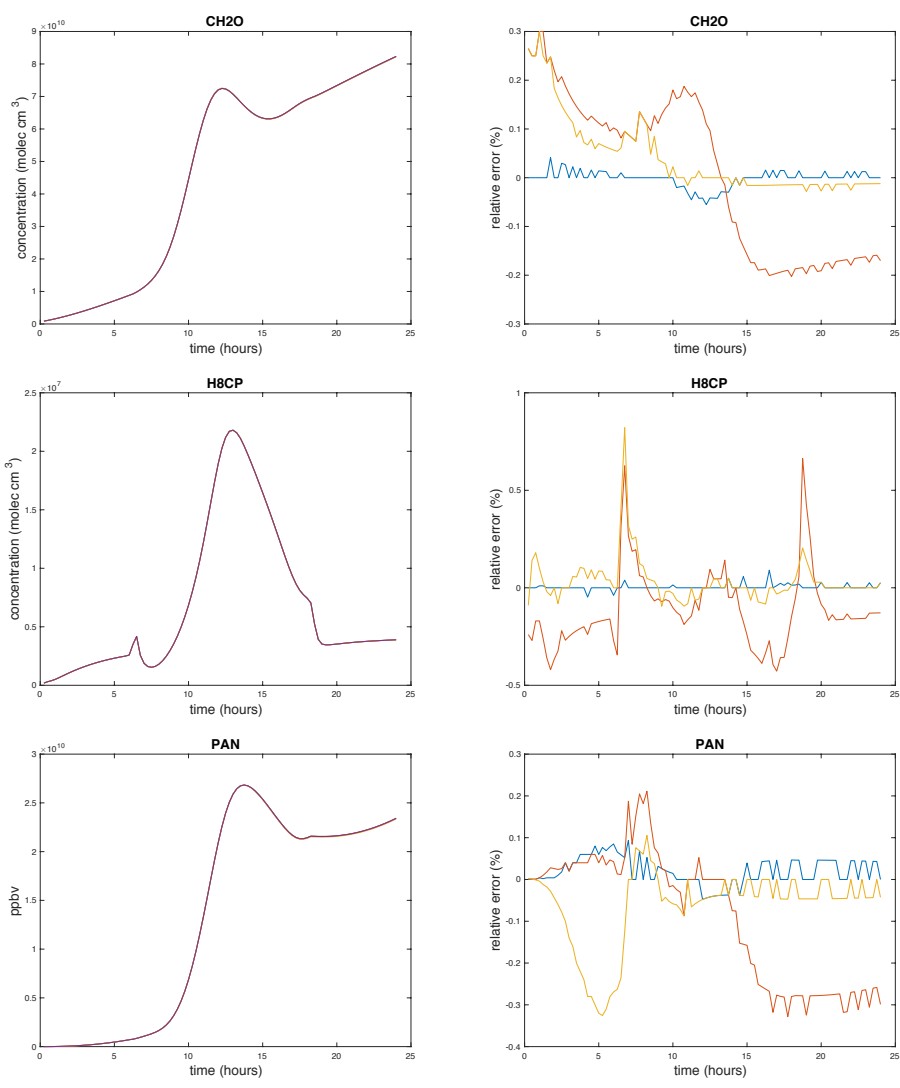

**Figure 2.** FLUX case. Same as figure 1 for $CH_2O$, H8CP and PAN.





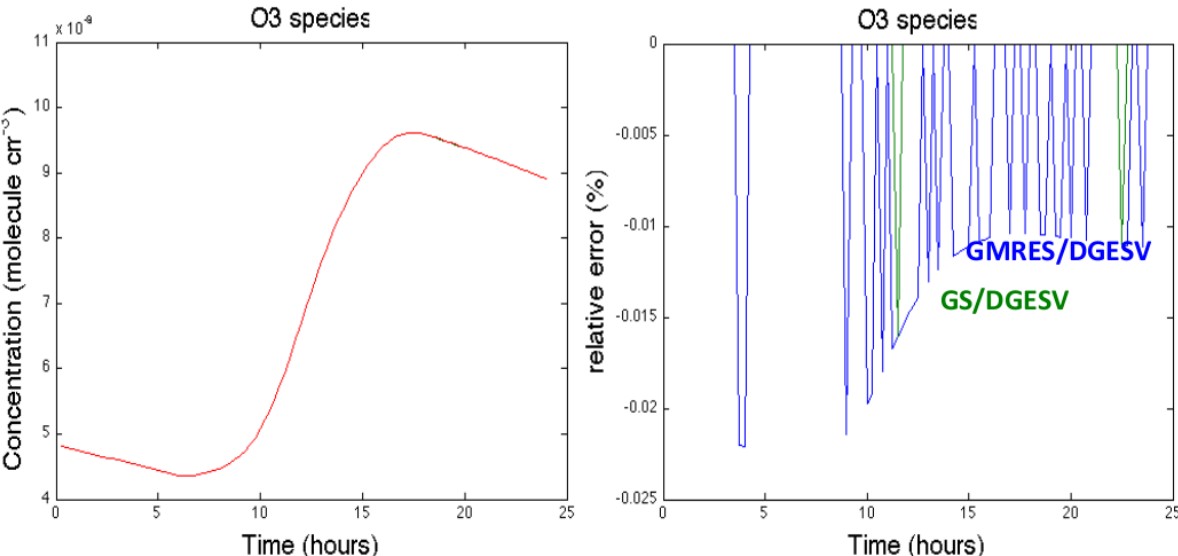

**Figure 3.** FLUX case. Time evolution of O$_3$ for the A2 experiment. Left panel: evolution of the concentration. Right panel: relative differences of the solutions using the iterative Gauss-Seidel (green) and GMRES (blue) algorithms compared to the direct method DGESV to solve the linear systems associated to the ASIS solver.

lived radical species have the largest variations. Those transition situations are the most difficult because not only the accuracy must be maintained but spurious numerical oscillations must be avoided. ASIS performs well here and adapts its timestep automatically to reach the required accuracy. The numerical treatment adopted to calculate an approximation of the Jacobian (Eq. 9 with $\beta = 1$) contributes greatly to damp numerical oscillations without significant degradation of the accuracy of the solution.

Equally, the approximations in the Jacobian are efficient to prevent the development of negative mixing ratios. In the two cases FLUX and STRATO, we did not encounter any significant (larger than $ATOL$) negative values during the course of the simulation and all the concentrations remain positive at the end of the 15 minute intervals before the photodissociation rates are updated.





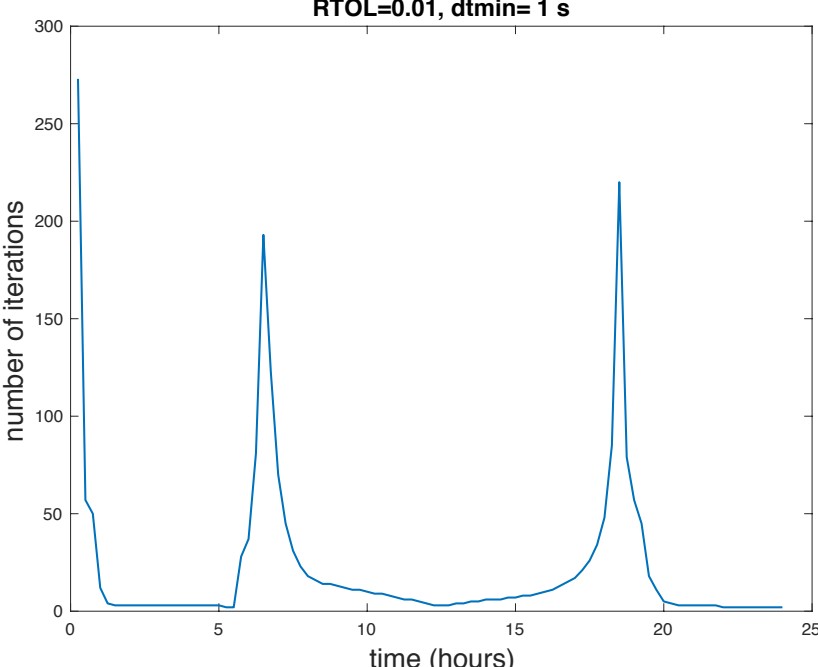

**Figure 4.** STRATO case. Number of timestep of the ASIS solver for each interval of 15 minutes in experiment A4.

In summary, the results of the two test cases confirm the properties searched in the design of ASIS. At the moderate accuracy required for atmospheric simulations the ASIS solver compares well with higher order schemes, and limits the computational cost while assuring mass conservation. The next sections illustrate how it performs in more realistic situations with implementations in state of the art global chemical transport models for Earth and Mars atmospheres.

## 4   Implementation within the MOCAGE model

For this study we have used the global version of the chemical transport model MOCAGE with an horizontal resolution of 2°x 2 °and 47 levels in the vertical from ground to 5 hPa. The chemical scheme is RACMOBUS, identical to the one used for the test cases of section 3. In addition to chemistry and transport by the large scale winds and by convection, the model includes the main processes that contribute to the sources and sinks of the species: surface emissions, scavenging by rain, dry and wet depositions. The timesteps for these processes is 15 minutes, photodissociation and chemical rate constants are updated at the same frequency. We report here simulations over three months from the beginning of August to the end of October 2011. Wind and temperature fields come from the operational weather analyses of the ECMWF. They are updated every 3 hours and linearly interpolated in between these time intervals.





**Figure 5.** STRATO case. Time evolution of selected species ($CH_2O$, $NO_2$, $NO_3$, $OH$) for the R2, and A4 experiments. The left column shows the mixing ratios or concentrations and the right one the differences in % for A4 relative to the R2 experiment.





The reference simulation, MR, uses the original solver for chemistry, an iterative semi-implicit scheme with assumptions of equilibrium for short lived species and species lumping for NOx and Clx families. The chemical timestep varies with altitude but is kept constant during the model integration. It increases from 20 s in the planetary boundary layer to 15 minutes in the stratosphere.

The simulation with the ASIS solver, MA, uses the same configuration for MOCAGE as MR except that the original chemical solver is replaced by ASIS with settings similar to experiment A3: RTOL= 0.025, ATOL= $10^4$ molecules $cm^{-3}$, a minimum timestep of $5s$, and the GMRES solver for the resolution of the linear systems.

The characteristics of the ASIS functioning implemented within MOCAGE can be first examined by the diagnostic of the number of sub-timesteps for chemistry. Figure 6 shows this number for 3 different levels for a date corresponding to the 15 th
of September at mid-day. In the mid-stratosphere, at 50 hPa, the number of sub-timesteps varies in accordance with what was found for the STRATO test case. At mid day or night the chemical system is in quasi steady state and this number is small, below 3. Close to the terminators this number increases up to 40-60 highlighting the change of regime of the chemical system when the photodissociation is activated or deactivated. During these transition phases the stiffness of the system increases and the sub-timesteps decrease to maintain the required accuracy. Also barely noticeable is an increase of the number of sub-
timesteps over the Antarctica coast at the edge of the polar vortex. In these regions the heterogeneous reactions acting at the surface of polar clouds are activated introducing disequilibrium of the concentrations of the chlorine species. It leads to a reduction of the sub-timesteps to cope with the rapid variations of the chemical composition of the air masses.

In the middle troposphere the same behavior is encountered near the terminators with a tendency to maintain reduced sub-timesteps during longer periods after sunrise or before sunset (figure 6). An increase of the number of sub-timesteps is also
encountered over the african continent at low latitudes. Those regions are prone to convective activity and injection of species by convection is activated leaving air masses far from chemical steady-state. Since the chemical evolution of the species is calculated after the transport processes, ASIS starts with a situation far from a chemical equilibrium and the number of sub-timesteps increases.

At the surface, figure 6 shows the same characteristics as in the mid-troposphere with an increase of the number of sub-
timesteps at the terminators and over the continents. Over the continents the surface emissions play a larger role than convection to destabilize the chemical system. Within MOCAGE the emissions are calculated according to inventories and deposited in the boundary layer. This is treated as an isolated process that changes the concentrations. As a result ASIS starts with situations out of chemical equilibrium and adopts small sub-timesteps, about 20 s compared to 60 s over the oceans.

Except for noticeable cases that are discussed hereafter, the species distributions of the MA simulation are close to those
obtained in MR. As an illustration figure 7 shows the zonally averaged distributions of $O_3, CO, OH$ and $HNO_3$ for the month of September. In most altitudes the differences are below 10 % with the largest differences in SH high latitudes in the lower stratosphere. Similar differences are found for the other species except for the $NO_x$ species in the lower troposphere and the chlorine species in the high latitudes of the SH during the formation of the stratospheric ozone hole.

In the lower troposphere examination of the code of the MR simulation reveals that approximations and steady-state as-
sumptions are made for the computation of the night-time $NO_2/NO_3/N_2O_5$ system. These approximations are valid in the





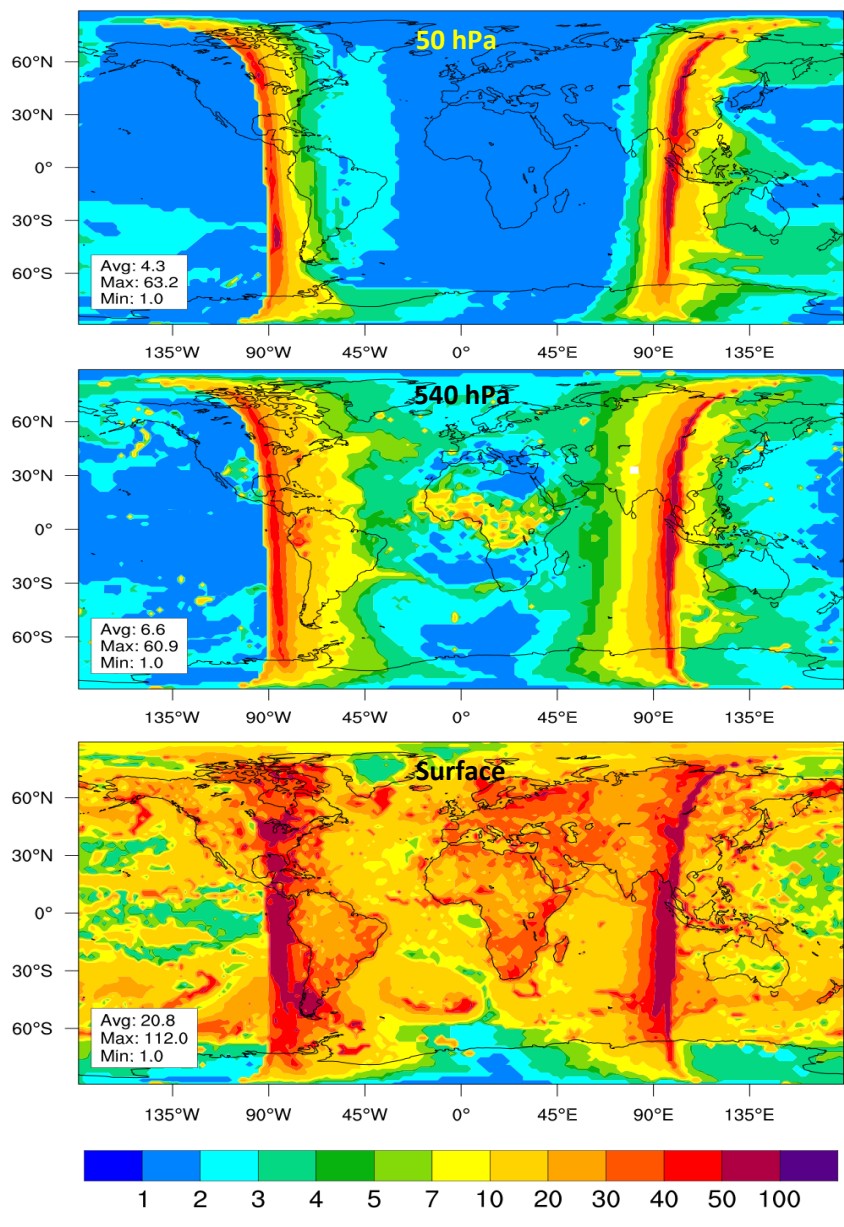

**Figure 6.** Number of sub-timesteps per timestep of 15 minutes in the MA simulation for the 15 th of September mid-day. 3 levels are presented representative of the stratosphere (50 hPa) , the mid-troposphere (540 hPa) and the surface.



stratosphere but fail in the lower troposphere where the pressure and temperature are larger. As a result the MR simulation under-evaluates the concentrations of $NO_3$ and $N_2O_5$.

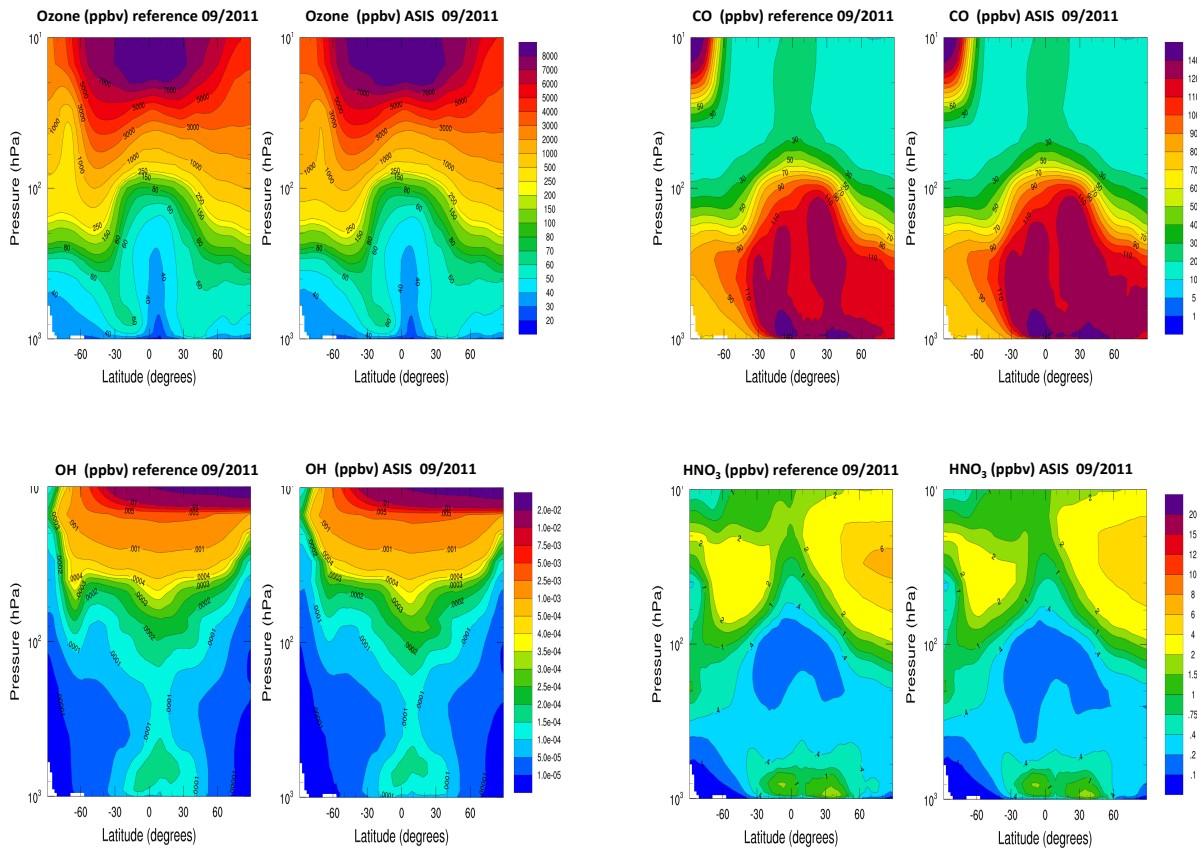

**Figure 7.** Zonal mean distributions of $O_3$, $CO$, $OH$ and $HNO_3$ for the month of September. The left columns shows results for the reference simulation, MR, the right column shows results of the MA simulation with the use of the ASIS solver.

Figure 9 shows for example the distributions of $NO_2$, $HNO_3$ and $N_2O_5$ at the lower level near the surface averaged over the month of October. In the region of surface emissions the MR simulation strongly underestimates the $N_2O_5$ concentrations.
5   Since the chemical scheme adopted for the present simulations does not include the formation of $HNO_3$ by the hydrolysis reaction of $N_2O_5$ on aerosols surface (Dentener and Crutzen, 1993), it has not a major influence on the other species. Nevertheless, the maximum values for $HNO_3$ and $NO_2$ are larger in the MA simulation than in the MR simulation.

Another significant difference between MR and MA is found in the simulation of the $ClO_x$ system in the lower stratosphere at high SH latitudes. In late August and early September the solar radiation comes back at high latitudes and the lower strato-
10   spheric $O_3$ is destroyed by catalytic cycles involving chlorine radicals (Solomon, 1999). The chlorine radical concentrations





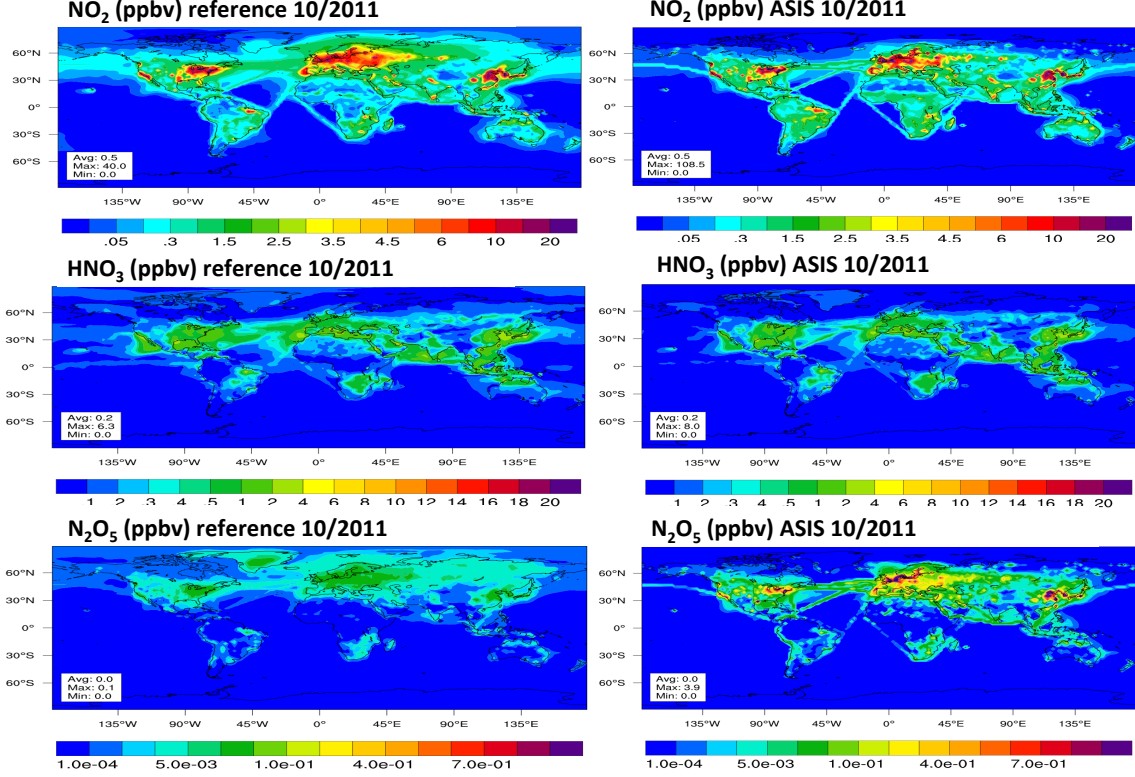

**Figure 8.** Monthly mean distributions of $NO_2$, $HNO_3$ and $N_2O_5$ at the surface for the month of October after a two month integration. The left column shows results of the reference simulation, MR, the right column shows results of the MA simulation with the use of the ASIS solver. The MR simulation under-evaluates the $N_2O_5$ in the lower troposphere.

are enhanced by the heterogeneous reactions on PSC's surface that convert HCl and $ClONO_2$ into $Cl_2$ that is photodissociated to form the chlorine radicals. In addition, the catalytic destruction of $O_3$ involves also the bromine species.

In the air masses prone to heterogeneous reactions on PSC the composition changes rapidly at sunrise and non linear processes, like the formation of $Cl_2O_2$, a key species for the $O_3$ destruction, play a major role. As a result the chemical system is very stiff and the ASIS solver diminishes the chemical timestep to a few seconds to maintain good accuracy. In these transient situations the original code in MR does not change its settings and a fix timestep of 15 minutes is used.

It results in the MR simulation showing a much more pronounced ozone depletion over Antarctica than the MA simulation. MR calculates ozone column contents as low as 100 Dobson Units (DU) whereas the MA simulation maintains values in the range of 150 DU. This is well illustrated in figure 9 that shows the evolution of the total ozone columns over two Antarctic





stations, Dumont d'Urville and Dôme C. For these 2 stations the measurements done by SAOZ instruments (Pommereau et al., 1988) at sunrise and sunset are also presented (data available at http://saoz.obs.uvsq.fr/).

Starting around 220 julian day the MR and MA simulations start to diverge. Over Dumont d'Urville, the station that sees first the return of the sunlight, the ozone decrease is about 50 % larger in the MR simulation than for MA. By day 260 the ozone column is just above 150 DU whereas it is in the 200 DU range in the MA simulation. Clearly the MA simulation is in better agreement with the SAOZ measurements.

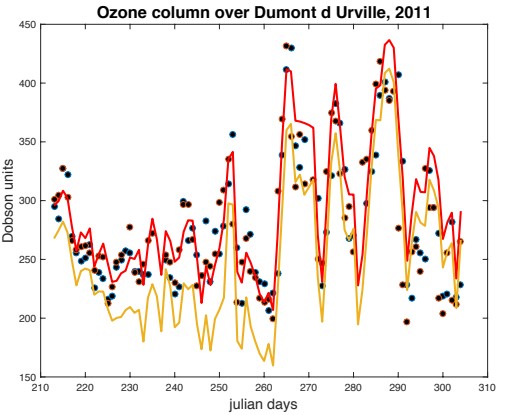
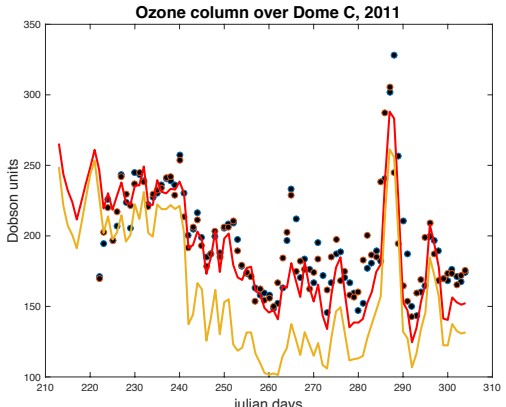

**Figure 9.** Evolution of the total ozone column over the Dumont d'Urville and Dome C antarctic stations. The dots are the observations of the SAOZ instrument, the orange line is the evolution calculated in the reference simulation, MR, and the red line the same output from the simulation MA using the ASIS solver.

The same behavior is seen for the Dôme C station. The ozone depletion starts slightly later, around day 240. In the MR simulation the depletion is very pronounced and the ozone column diminishes rapidly in a few days from 240 to 150 DU, and further decreases at a slower rate to reach a minimum of 100 DU at day 260. The MA simulation shows a more continuous decrease from day 240 to 260 with an ozone column reaching a minimum of 150 DU. The MA simulation is here again in very good agreement with the SAOZ observations.

Implementation of the ASIS solver within MOCAGE has thus revealed two weaknesses of the original model. One problem is in a limitation on the validity of assumptions made to compute the night-time distribution of the NOx species. It can be solved by adequate coding. The other one is a lack of accuracy in the resolution of the chemical system in specific situations in the lower stratosphere. This can certainly be avoided by a drastic reduction of the timestep, but it would need the implementation of a time varying timestep strategy somewhat similar to the one adopted for ASIS.

Clearly the implementation of ASIS within MOCAGE is very beneficial to the model simulations and increases the confidence on the model results. In addition, further evolution of the model with adoption of different chemical schemes or addition of new reactions is very easy with ASIS.



There is however a price to pay in terms of computer time. Overall the MA simulation takes 4,7 times more computational time than the MR simulation. This number could be certainly decreased by further tuning of the parameters of the solver, $RTOL, ATOL$ and $\delta t_{min}$, and maybe also by the use of the Gauss-Seidel algorithm instead of GMRES in situations where the solution of the linear system converges easily.

Our experience with ASIS shows that since various processes are computed by a series of operators the solver starts new timesteps with situations often out of chemical equilibrium and must use small sub-timesteps. To alleviate this, one possibility is that tendencies from these operators are computed and stored rather than used to update the species concentrations. The tendencies can then be used to solve the system though their introduction in the term $F$ of Eq. (7). We have tested this option for the species emissions at the surface and found that the number of sub-timesteps is decreased by a factor 2 in the lower

troposphere. It remains to be seen if other processes can be treated that way. Emissions are the most straightforward because the resulting tendencies are positive and cannot lead to the calculation of negative concentrations.

Another issue lies in the parallelisation of the computations. In the reference simulation the computational cost is equal for each grid-point at a given level and good parallelisation is obtained with an equally spaced latitudinal band decomposition (and use of openMP directives). When ASIS is used the computational cost in each grid-point depends on the state of the chemical

system. As illustrated in figure 6, in the stratosphere and upper troposphere more computer time is needed near the terminators and in case of PSCs induced chemistry. In the lower troposphere more computer time is spent in grid-points influenced by surface emissions, and convective and boundary layer transport processes. A speedup of 15 was however obtained for the MA simulation on our cluster computer (using one node and 16 cores of our BULL computer) with a decomposition that groups more longitudes in the SH than in the NH near the poles. But further tuning would be required if more nodes are to be used. This

tuning could vary with season and additional parallelisation could be introduced with domain decomposition on the vertical.

## 5   Implementation within the LMD Mars model

To illustrate the versatility of the ASIS solver, we present results of the implementation of ASIS in the LMD Mars model with photochemistry (Lefèvre et al., 2004). This Mars GCM describes the evolution of 19 species (table 5) by means of 54 chemical or photolytic reactions. The bulk atmosphere of Mars is composed of 95 % of $CO_2$ with trace amounts of $H_2O$. As a result, the

only processes that initiate Martian photochemistry are the photolysis of $CO_2$ and $H_2O$ by ultraviolet solar light. Therefore, the photochemistry of the lower atmosphere of Mars can be summarized by the interactions between the oxygenated species $O(^1D)$, O, $O_3$ produced by $CO_2$ photolysis and the hydrogen radicals H, OH, and $HO_2$ produced by $H_2O$ photolysis. These processes are similar to those occurring in the Earth's mesosphere, with comparable conditions of pressure and temperature.

In the standard version described in Lefèvre at al. (2004), the LMD GCM with photochemistry uses the Euler backward

method (EB) expressed in (3) to solve its chemical system. As mentioned earlier, this method is positive, stable, and can be computationally effective but does not maintain mass conservation. Iterative evaluations of $C^{t+1}$ are performed in the lower atmosphere of Mars to reduce this problem. In the Mars thermosphere, another option is used in the LMD model which consists in shortening the timestep $\delta t$ according to the species with the shortest lifetime (González-Galindo et al., 2009). In both cases,





**Table 5.** List of species used in the Mars model simulations.

| |
|---|
| $O(^1D), O(^3P), O_2, O_3,$ |
| $N, N_2, NO, NO_2, N(^2D),$ |
| $H_2, H_2O(gas\&solid), H, OH, HO_2, H_2O_2,$ |
| $CO, CO_2, Ar$ |

species lumping and assumptions of photochemical equilibrium are used to increase accuracy and avoid very small timesteps. However, conditions of photochemical equilibrium change at night and are also very dependent on altitude. For instance, on Mars, the lifetimes of $O(^3P)$ and H vary between less than 1 second near the surface to several years at 100 km. Such stark variation prevents assuming photochemical equilibrium or using Eq.(3) throughout the atmosphere. Thus, despite its apparent

simplicity, the Euler backward method may complicate the problem by requiring different treatments for specific species or specific parts of the atmosphere.

Figure 10 compares the results obtained with the Euler-Backward (EB) and ASIS solvers applied to a box-model version of the LMD Mars model. The atmospheric pressure/temperature is 5.4 hPa/212 K at the surface and 0.2 hPa/140 K at 30 km. In both cases the integration starts at noon, and stops after one Martian solar day of 24h40 mn. The photodissociation rates are

calculated every 15 minutes using the TUV radiation model adapted to Mars. The timestep of the EB solver is fixed to $\delta t = 7.5$ mn as done in the Mars GCM. ASIS uses the variable step size strategy described in section 2.3, bounded by a maximum value of 15 mn and the minimum timestep of 10 s. RTOL is fixed to 0.05 and the ATOL density is equivalent to a mixing ratio of 10 pptv. ATOL is therefore variable with altitude. The resolution of the linear systems associated to ASIS in done using the DGESV direct solver. We found that these settings were adequate to reach a satisfying compromise between accuracy and

computing time.

At the surface, figure 10 shows that the ASIS solver calculates an $O_3$ mixing ratio that is lower by 3 to 6 % compared to the EB solver. This difference is related to the lack of accuracy in the treatment of the HOx species in the EB solver, which assumes that OH and $HO_2$ are at photochemical equilibrium at all times within the HOx family. This assumption is close to reality during the day, but becomes problematic at sunrise and sunset and is wrong at night, when the $HO_2$ lifetime can reach

several hours at the surface. As a result, the OH mixing ratio calculated by the EB solver is overestimated by a factor of 10 compared to ASIS, which does not require any a priori assumption on chemical lifetimes and provides an accurate solution throughout sunset and nighttime. At sunrise and sunset ASIS reduces the chemical timestep down to 10 s to solve the sharp transitions in the concentrations of short lived-species H, OH, O and NO. Outside these critical (but short) periods, the Martian settings of RTOL and ATOL allow timesteps that increase rapidly and may reach $\delta t = 15$ mn without sacrificing the accuracy.

Thus, in the example of figure 10, at the surface level, the number of chemical timesteps performed by ASIS over one Martian day is only 12 % larger than in the EB simulation.

The box-model simulations at 30 km are performed at the hygropause level where the production rate of HOx radicals by $H_2O$ photolysis is largest. This results in a maximum stiffness of the system at sunrise and sunset, when the $H_2O$ photolysis





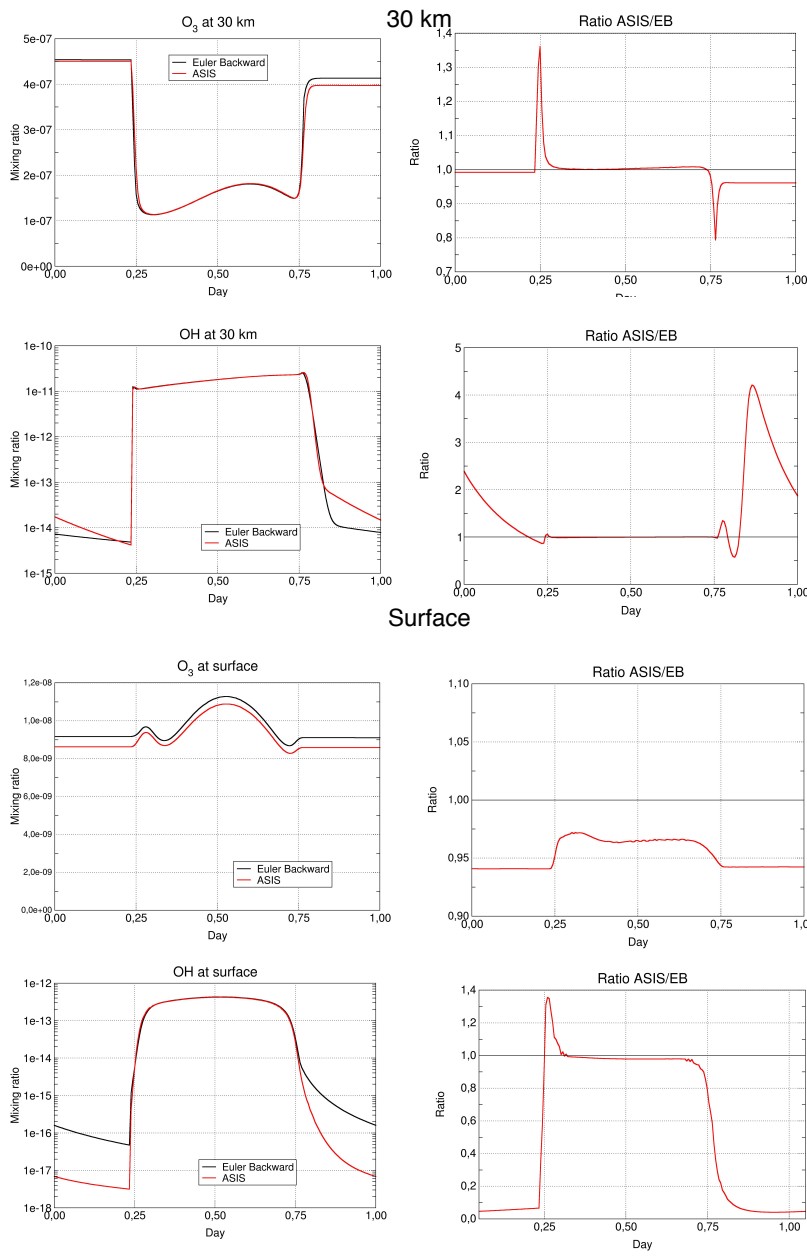

**Figure 10.** Comparison of the Euler-Backward (EB) and ASIS solvers applied to the Mars box-model version. The left column shows the mixing ratios of O$_3$ and OH and the right one the ratio between the ASIS and EB experiments. Results are presented at 30 km (top) and at the surface (bottom), for equatorial conditions in northern spring (solar longitude Ls = 70 $^\circ$). Local noon is at day 0.5.





rate varies rapidly. Those critical day/night transitions show large differences between the ASIS and the EB simulations. In the EB run, ozone is integrated implicitly by Eq.(3) at night and is assumed to be at photochemical equilibrium within the Ox family during the day. This abrupt change in treatment contrasts with the smooth transition carried out with the timestep adaptative scheme of ASIS. At the price of a strong reduction of the timestep to maintain the required accuracy, ASIS calculates

an $O_3$ mixing ratio that is respectively 35 % larger and 20 % smaller than in the EB run at sunrise and sunset. Both solvers give the same results during the day. However, the more accurate description of the $O_3$ increase at sunset by ASIS induces a 5 % difference with the EB solver that persists into the night. Regarding OH, the simulation at 30 km confirms the weakness of the steady-state approximation for HOx at night in the EB scheme. In ASIS, the stiffness of the system diagnosed by the solver remains high in the first hours following sunset (due to strong curvature of the solution for H, not shown here) and leads to a

reduction of the timestep to about 30s. The nighttime OH mixing ratio is larger by a factor 2 to 4 than in the EB simulation. For this extreme case of stiffness in the Mars atmospheric chemistry, the total number of chemical timesteps executed by ASIS over one Martian day is 65 % larger than in the EB simulation.

In its three-dimensional implementation, ASIS is called by the LMD GCM at each physical timestep $\Delta t = 15$ mn. The ASIS settings in the GCM are identical to those of the box model presented earlier, i.e. the solver may select any sub-timestep value

between $\Delta t$ and the minimum value $\delta t_{min} = 10$s. To compare the GCM performances with ASIS and with the EB method, two simulations of 150 Martian solar days have been performed with each method starting with an identical initial situation.

Figure 11 shows the number of sub-timesteps per physical timestep of 15 mn in a GCM simulation of northern spring (Ls = 70 °) using ASIS. For the three levels presented here (surface, 30 km, 80 km) the number of sub-timesteps is equal to 1 or 2 for a large fraction of time. This is the case when the chemical system is in equilibrium, far from the terminators at night or

during the day. As in the MOCAGE model, at the terminators the number of sub-timesteps increases dramatically to cope with the change of chemical regime at the day-night transitions. The maximum number (40-50) is found at sunrise at 30 km and is essentially driven by the abrupt changes in OH and $O_3$ already seen in figure 10. At the surface, an increase in the number of sub-timesteps is also visible near the North pole. This is related to fast heterogeneous reactions of HOx species on water-ice clouds (Lefèvre et al., 2008), a process similar to that occurring with chlorine on Earth stratospheric clouds. In those cases

ASIS adopts a smaller timestep to resolve with good accuracy a system that is locally away from chemical equilibrium.

Figure 12 compares at 30 km the results of GCM simulations using either the EB or the ASIS solver. Both schemes give distributions of $O_3$ and OH that are in general very close during daytime and away from the terminators. At the terminators, ASIS calculates $O_3$ amounts that are about 50 % larger than EB at sunrise and 25 % smaller at sunset. These large differences are similar to those found with the box-model runs (figure 10) but are limited in time and space. However, the better description

of $O_3$ by ASIS across the terminators may be crucial when comparing the GCM to Martian ozone measurements performed at the terminators by the solar occultation technique. Regarding OH, the GCM results confirm the poor description of the HOx chemistry by the EB scheme at the terminators and especially at night, with values 4 times smaller than with ASIS. The amount of nighttime OH is small relative to daytime values. Thus, the bias in the EB scheme does not affect significantly the oxidizing capacity of Mars simulated by the GCM. Nevertheless, similarly to ozone, the more accurate description of OH and



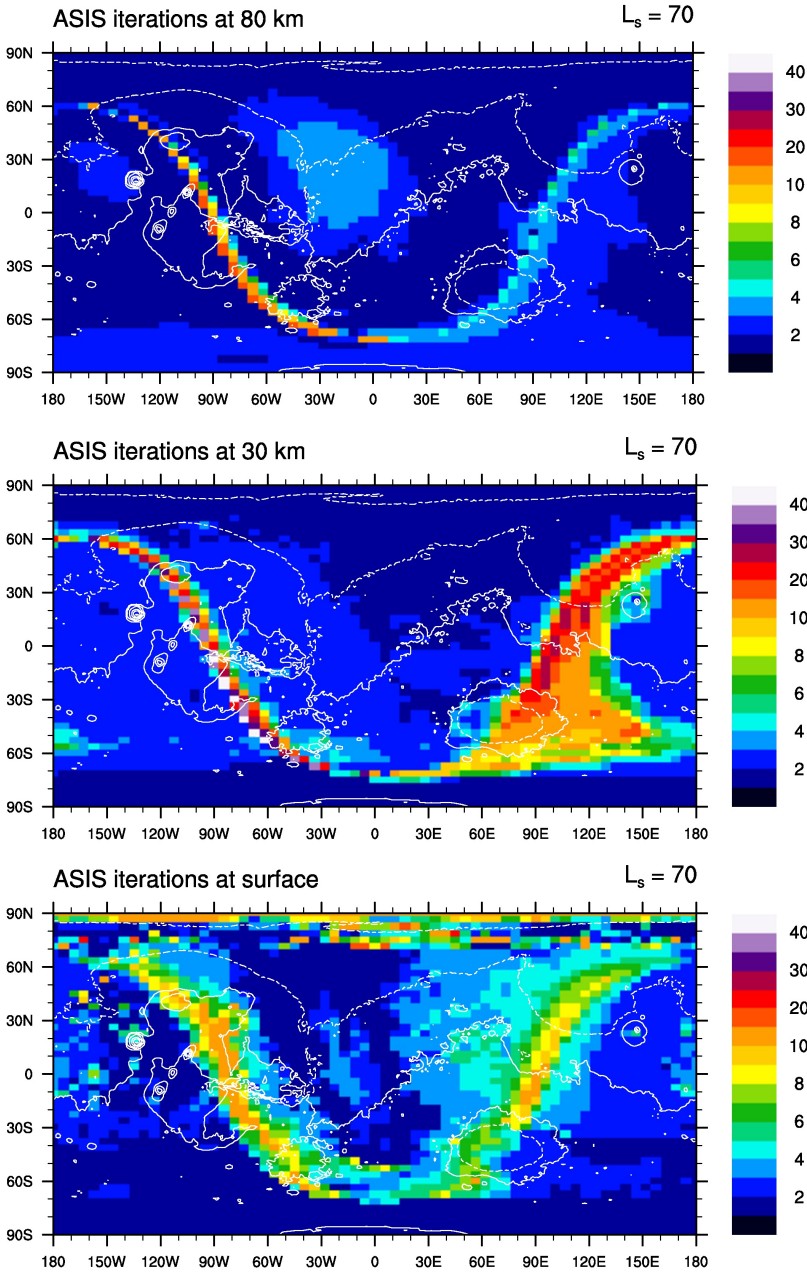

**Figure 11.** Number of sub-timesteps per time interval of 15 minutes in the LMD Mars GCM in northern spring (solar longitude Ls = 70 °). Three altitude levels are represented at 80 km (top), 30 km (middle), and the surface (bottom). Local noon is located at longitude zero. The white contour represents topography, with a 4-km interval.





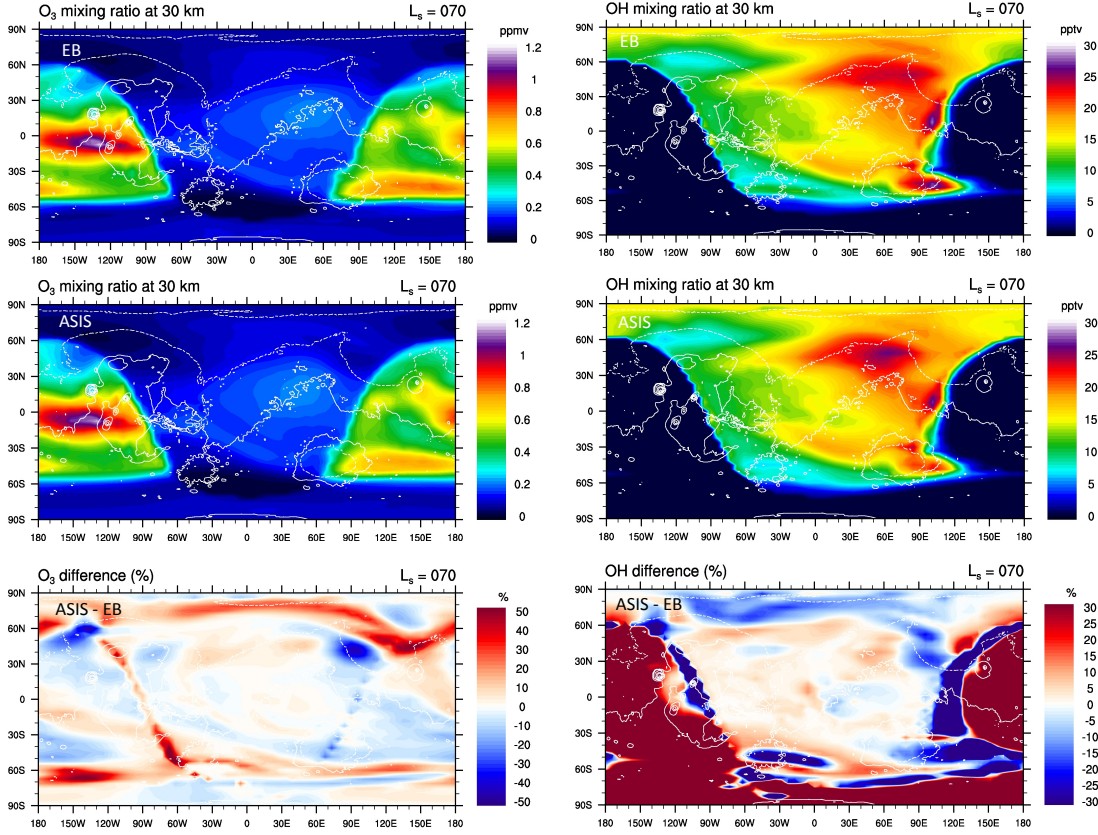

**Figure 12.** Distribution of $O_3$ (left) and OH (right) at 30 km calculated by the LMD Mars GCM in northern spring (solar longitude Ls = 70 °). Top : Euler-Backward (EB) solver. Middle : ASIS solver. Bottom : relative difference (%) between ASIS and EB, using thresholds of 10 ppbv for $O_3$ and 10 pptv for OH. Local noon is located at longitude zero. The white contour represents topography, with a 4-km interval.

the Martian nighttime chemistry in general is an important advantage brought by ASIS for the interpretation of the numerous observations of nightglow or measurements by stellar occultation carried out on that planet.

## 6 Conclusions

The ASIS solver has been designed to cope with the various situations encountered within the numerical simulation of the atmospheric chemistry. The main properties of the solver are mass conservation, an approximation of the Jacobian matrix of the chemical fluxes that stabilizes the associated system of differential equations, a time stepping varying module to control accuracy, and a code implementation that allows an easy adaptation to various chemical schemes. In box model test cases,



the numerical solutions obtained with the ASIS solver were found in good agreement with those of multi-step algorithms like Rosenbrock's and Gear's methods.

The ASIS solver has been implemented in two 3D models of the Earth (MOCAGE) and Mars (LMD model) planets. The results with MOCAGE using ASIS reveals two weaknesses of the original semi-implicit solver. One is related to the calculation
of the partionning of the NOx species at the surface and the other to an overestimation of the ozone depletion in the Antarctic stratospheric vortex in Spring. In the simulation of the Mars atmosphere ASIS gives more accurate simulations during day-night transitions and at night for the HOx species. These results stress the importance of having accurate enough numerical solutions, otherwise differences between model simulations and observations could be wrongly attributed to missing chemistry or misrepresentation of some physical processes.

The model simulations show the benefit of using a chemical solver with good properties such as mass conservation and controlled accuracy. This objective can be achieved using multi-step high order algorithms but the computational cost of those schemes increases rapidly with the number of species considered. Since ASIS is implicit and one step, a single linear system has to be solved for each iteration. For this, direct or iterative algorithms can be used. The direct methods based on LU decomposition see their computational cost increasing at least quadratically with the number of species, whereas the cost of
iterative solvers increases rather linearly. Within ASIS we found that the GMRES iterative algorithm is stable and efficient, and is competitive in terms of CPU cost compared to the direct DGESV algorithm.

In atmospheric models the computational cost is a key issue and parallelisation of the computations must be efficient to reduce the elapse time spent for the simulations. As pointed out earlier the amount of computation spent by ASIS to solve the chemical system can vary significantly from one grid point to another. This renders the equilibrium of tasks more difficult if a
domain decomposition strategy is adopted to implement the parallelisation. As already discussed with the surface emissions, one possibility to diminish the number of iterations and the heterogeneity in the CPU used at each grid-point is to account for non chemical tendencies in the species continuity equations (term $F$ of Eq.4). Rather than updating the concentrations after each process the resulting tendencies could be added and integrated within ASIS. This strategy has been adopted for example by Menut et al. (2013) for the CHIMERE model, it remains to be seen if the stability and the positivity of the solution can be
maintained.

The present version of the ASIS solver adresses the evolution of the concentrations in gas phase only. For some applications the aqueous phase associated with the presence of clouds must be also considered (e.g. Leriche et al. 2013). The chemistry module has to solve both gaseous and aqueous phases chemistry as well as mass transfer reactions between gas and liquid phases. There is a priori no difficulty to add the prognostic concentrations in the water phase to the system of equations and
make a linearization similar to what is done in Eq. (6). However, addition of aqueous reactions tend to increase the stiffness of the numerical ODE (Audiffren et al., 1998) so the performances of ASIS could diminish and may result in reduced timesteps and increased computer time.

In conclusion, the ASIS solver can deal with many situations encountered in modeling atmospheric chemistry for a computational cost affordable by CTMs and GCMs that include comprehensive chemical schemes. Evolution of ASIS solver to treat
aqueous phase chemistry is planned in the near future.





## 7 Code availability and details on code implementation

The Fortran code to run the ASIS solver on the FLUX case described is section 3 is available from CERFACS. Requests to access the code can be addressed to D. Cariolle (cariolle@cerfacs.fr). The ASIS code is property of the CERFACS and includes libraries that belong to other holders.

The code associated to the chemistry model includes subroutines that define the mechanism and those more specific to the ASIS solver. At this stage we have not developed an external driver or a pre-processor that would generate specific codes based on the adopted mechanism. This choice was done because our experience is that the maintenance of the driver outputs can be somewhat cumbersome when many developers work in parallel on a CTM. In addition, the code generated by the driver must be often optimized for the computer used and adapted to the CTM. It is therefore not used directly, which introduces further

constraints on the maintenance of the overall code.

Our approach is rather to define the mechanism by a limited number of fortran subroutines that are simply added to the other routines of the code. The $num\_species$ routine names and numbers the species, the $indices\_reactions$ routine does the same for the reactions. The reactions are classified in 3 groups:

1/ A —> b B + c C

2/ A + A —> b B + c C

3/ a A + b B —> c C + d D

The first group includes photodissociations and thermal decomposition of the species. This classification is done in order to optimize the calculation of the terms of the matrix $M$ of Eq. (7). Some reactions gives more than 2 products and fractional sub-reactions must be introduced. For example the following reaction with fractional products:

$HC5P + NO3 --> 0.021 * HCHO + 0.239 * ALD + 0.828 * KET + 0.699 * HO2 + 0.040 * MO2 + 0.262 * ETHP + 0.391 * XO2 + NO2$

will be decomposed in 4 sub-reactions within the $indices\_reactions$ routine:

$zloc2 = Z4SPEC(1.0, JPHC5P, 1.0, JPNO3, 0.021, JPHCHO, 0.239, JPALD)$

$Zindice\_4(JP4\_HC5P\_NO3\_i) = zloc2$

$zloc2 = Z4SPEC(0.0, JPHC5P, 0.0, JPNO3, 0.828, JPKET, 0.699, JPHO2)$

$Zindice\_4(JP4\_HC5P\_NO3\_ii) = zloc2$

$zloc2 = Z4SPEC(0.0, JPHC5P, 0.0, JPNO3, 0.040, JPMO2, 0.262, JPETHP)$

$Zindice\_4(JP4\_HC5P\_NO3\_iii) = zloc2$

$zloc2 = Z4SPEC(0.0, JPHC5P, 0.0, JPNO3, 0.391, JPXO2, 1.0, JPNO2)$

$Zindice\_4(JP4\_HC5P\_NO3\_iv) = zloc2$

paying attention not to duplicate associated fluxes.

Once the definition of species and reactions is completed, the calculation of the matrices (Eq. 7) is done by the $fill\_matrix$ routine, the timesteps are monitored by the $define\_dt$ routine and the resolution of the linear systems by the $Solvesys$ routine.



*Acknowledgements.* This work was supported by the Monitoring Atmospheric Composition and Climate (http://www.gmes-atmosphere.eu/) and COPERNICUS Atmospheric Monitoring Service (https://atmosphere.copernicus.eu/) E.U. projects. The authors thank Emanuele Emili, Andrea Piacentini and Michael Prather for their very helpful comments on the manuscript.



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
