# Peer review of "ASIS v1.0: an adaptive solver for the simulation of atmospheric chemistry"

_Geoscientific Model Development, 2016_

## Referee Comment (RC1) · Anonymous Referee #1 · 16 Jan 2017

General comments and recommendation.

The authors of this paper describe a newly developed chemistry solver. While reading the paper I became convinced that the ASIS solver is working well. The trick with the Delta factor, eq. 9, is an interesting way to cope with the large range of lifetimes of the tracers in an automatic way. ASIS is very flexible and does not need explicit equilibrium solutions of fast reactions. I can agree with the authors that the high accuracy of solvers like Rosenbrock is often not needed for CTM simulations. The examples (box model, MOCAGE, Mars model) are very instructive and clearly show the properties of ASIS. As such I am in favour of publishing this work in GMD.

However, there are a few major comments and a couple of more minor points which I would suggest the authors address before the paper is published.

First, from the paper it is not so clear why a new solver is needed. In fact, in the paper there are comments which may make the CTM modeller decide to stick to solvers like Rosenbrock. In particular I would like to see a more detailed comparison against Rosenbrock concerning run time and accuracy, for small and large numbers of chemical species. What are the main reasons to replace Rosenbrock by ASIS? The abstract could be extended also in this direction.

Some comments on the advantage of ASIS I found a bit misleading, e.g. on p15: "At the moderate accuracy required for atmospheric simulations the ASIS solver compares well with higher order schemes, and limits the computational cost while assuring mass conservation", but this improved run time is linked to lower accuracy. For the same accuracy I got the impression that run times were comparable.

I would expect to see more references (especially about chemical solvers and their properties) in the introduction.

More specialised comments:

- Title: "adaptative" -> "adaptive"

- Abstract: From the abstract it is not clear what the advantages of ASIS are with respect to the Rosenbrock and Gear solvers. Why do we need an other solver?

- Introduction: There are no references given in the introduction to the general literature on chemical or differential equation solvers. An introduction should sketch the starting point of the work - the state-of-the-art - and in this way clarify how the new developments described in the paper advance this present knowledge and models. I suggest that the authors add a section with references discussing the current status concerning solvers in relation to chemistry models. Several references are provided later on in the paper, but the current list is not very extended and could be expanded somewhat.

- p2, l3: "ODE" is not defined

- At several places: I suggest to replace "resolution" by "solution"

[Figure]

- p2, l12: "... for its resolution". I suggest to remove these words, or replace by something like "... to achieve high accuracy".

- p2, l12: "Adequate algorithms must then be used for its resolution". I would expect a review of the literature on this topic at this location.

- p2: The four properties "conservation of mass", "accuracy", "positivity", "flexibility" are generally considered in the context of solvers. Again, I would like to see a set of relevant literature references, and a brief overview of the available solvers, or classes of solvers.

- Introduction: What is the main motivation for developing a new solver? There are already a number of solvers readily available to choose from, which also satisfy a number of the properties mentioned. What are disadvantages of these other solvers, e.g. Rosenbrock, EB, twostep ?

- p3, l25: The second term on the left is diagonal. Please explain why ? Diagonal in which space?

- p3, l27: "mass conservation is not maintained". Can you add a reference or text book? Is it possible to describe in one line how the reader may understand that there is no conservation of mass?

- p4, l8: Please name a few solvers that use this approach.

- p5, eq9: I guess an other requirement is delta_l,m = 1 - delta_m,l, which is fulfilled.

- p5, l27: Why is beta >= 1 required? Should this be beta > 0?

- p8, l6: if -> is

- p9, p10: Are the tables 1, 2, 3 really needed for this paper? Ref to Crassier and to the MOCAGE model description may be enough. The paper is already a bit long.

- p11, l2: "the ASIS solver gives acceptable solutions with less computation than the

higher order schemes." What the authors show is that the A1, which has comparable accuracy as R1 and G1, needs a smaller time step and more, but less complicated computations. In the end the effort is similar for the same accuracy. I do not think it is fair to claim "less computations than the higher order schemes", because this is at the expense of accuracy. Is it possible to relax the tolerance in the R1 and G1 schemes and to speed up these schemes in this way as well? Would this lead to similar performances as A2 or A3?

- Fig. 2: What is "H8CP" ? Also explain "PAN".

- P14, l4: damp -> use e.g. dampen, reduce, or diminish

- Fig.6. The colour scale is unclear: Does the colour between 1 and 2 mean there are two substeps or one substep? I assume "number of sub-timesteps" can only take integer values.

- p17, l16: "disequilibrium" Replace by something like "out of equilibrium"

- p19, l3: Replace "Figure 9" by "Figure 8"

- I find the discussion of Figure 9 a bit long. It is good to mention the large time step reduction in ASIS, and the figures speak for themselves.

- p22, l8: (Use of tendencies in the term F) I believe this approach has been used by e.g. Chimere. Adding emissions during the chemistry solver time steps has been used by more models. Please refer to this.

- p22, l29: Please add a reference for Euler backward.

- p23, l11: "mn" I assume you mean "minute"?

- p23, l13: ".. to ASIS in done .."

- p25, l32: "4 times smaller". How can I see this from the figure 12, which has a scale ranging between -30 to 30%?

- p28, "This objective can be achieved using multi-step high order algorithms but the computational cost of those schemes increases rapidly with the number of species considered." It would be useful if the authors can make this point more explicit. How does the computational cost of ASIS compare with Rosenbrock as a function of the number of species. Perhaps the authors could introduce a figure to demonstrate this dependence.

- p28: I was wondering if ASIS could be used for adjoint (4D-Var) type of applications?

- p29, code availability: "The ASIS code is property of the CERFACS and includes libraries that belong to other holders." Does this imply any restrictions if other parties want to use the code? Is there a kind of license for using the code?

—————————————————

---

## Referee Comment (RC2) · Anonymous Referee #2 · 16 Jan 2017

The manuscript of Cariolle et al. describes the development and testing of a chemical solver suitable for implementation in models of atmospheric chemistry (ASIS). It appears that this solver is not intended to be available to the community due to the restrictive nature of the "code availability" (Section 7). Rather, it appears that the authors have chosen for their own reasons to develop an in-house solution instead of adopting an open source solver package such as KPP, and wish to use this publication to describe its basic features.

The formulation of the solver and its implementation are described at a level of detail appropriate for a paper in a technical journal such as Geoscientific Model Development. After describing the formulation of the scheme, a number of tests are described. The scheme is compared with standard reference schemes in a box model for two 24 hour test cases representative of the polluted urban boundary layer, and the mid

stratosphere, where the scheme is found to perform adequately. Further to describing these test results, the manuscript goes on to describe the results of 3 month simulations in global chemical transport models of Earth and Mars. The ASIS scheme is shown to improve the results of simulations in these CTMs due to its ability to adjust its sub-timestep based on the stiffness of the chemical system in each model grid cell. Two examples of improved performance are given for the simulation of the Earth's atmosphere: improved simulation of nighttime NOx chemistry in the boundary layer, and improved simulation of Antarctic ozone depletion in the stratosphere. These improvements come at the cost of increased model runtime; approximately 5 times more execution time is required compared to the simulations performed using the original chemical solver in the MOCAGE CTM. The authors speculate that various optimisations may help to reduce this performance penalty. It isn't clear to me though, why such an expensive solver should be preferred over similarly expensive solvers available through open source packages such as KPP which can provide comparable accuracy and runtimes. If the authors do not seriously intend to make their code generally available to the community, and are content with limiting its use to in-house applications, differentiating their product from other alternatives in this way is arguably outside the scope of the manuscript.

I find it a shame that only 3 month simulations are performed with the global models using the ASIS scheme. It would be very interesting to know what effect the replacement of the chemical solver would have on many other features of global atmospheric chemistry, such as the global oxidising capacity and related aspects such as the methane lifetime and tropospheric ozone budget. Such simulations would require much longer runs, of at least 12 months plus adequate spinup time. Perhaps the authors have avoided such longer simulations due to the high cost of running their model with the ASIS solver. While I regard the short length of the global test simulations as a shortcoming of the paper, I think that performing longer test simulations would be outside the scope of this study. I would just recommend that the authors do not consider this new solver to be completely evaluated until such comparisons have been performed.

In general the manuscript is clearly structured, but contains a large number of grammatical mistakes and non-idiomatic language. Somewhat surprisingly, I did not find that these language issues prevented clear understanding of the scientific and technical details being communicated. Nevertheless the manuscript would benefit from copy editing for correct English before being accepted into GMD.

―――――――――――――――――――

---

## Referee Comment (RC3) · Anonymous Referee #3 · 20 Jan 2017

This paper describes the implementation of the "Adaptive Semi-Implicit Scheme" (ASIS), a chemical solver for use in atmospheric simulations. The paper has a general discussion on the requirements on a chemical solver, before describing the ASIS formulation, and then giving some examples of its implementation in box models and GCMs.

I found the discussion informative, and the authors reasoning behind the formulation is well explained, along with the benefits that ASIS brings over other schemes. I would recommend this paper for publication, with some modifications.

**General Comments**

Due to the large number of tests performed it was a little difficult to keep track of the settings used in each case. I would recommend giving a master table (or tables) giving

the configuration for each shorthand used in the plots (e.g. R1, G1, A1, A2, A3, A4, MR, MA, EB, ASIS etc.) and what the settings are used for each (e.g. values for ATOL, RTOL, using ode23s, ode15s, DGESV, GS, GMRES etc.), the experiment run (e.g. FLUX, STRATO, MOCAGE, Mars Box Model, LMD Mars GCM etc.), and also the chemistry scheme used (e.g. RACMOBUS, Mars). This would be especially helpful for comparing between sections, as it can be difficult to pick out this mass of information from the text.

I would also like more discussion of computational cost - the timestep is discussed in detail, but is rarely then compared to the overall run-time. It is certainly interesting to see where the timestep changes, but in terms of usability it would be handy to know exactly how much more time it took. This could be included in e.g. table 4 for the section 3 cases, perhaps as a ratio relative to the R1 case. These numbers should also be clearly stated for all other cases.

The run length is also a bit short for most cases. The 1 day simulations for the box model are very short, especially when it appears that the A2 case is diverging from the R1/G1/A1 cases. Have these simulations been run for longer, and if so, how do the results of A2 (& A3) evolve? Also, A3 is not plotted at all, but these settings are then used for the MOCAGE simulations. The A3 results should also be included in the plots in Figures 1 & 2 (or plotted separately with a larger scale if required).

The GCM lengths are slightly better (3 months and 150 days), but still short. Are there plans to do longer runs with a full evaluation and budget analysis? The results presented here have highlighted deficiencies in the existing models, but a full analysis on longer simulations would be required to properly validate ASIS, as opposed to this paper which describes its implementation.

I do especially like that the authors have tested ASIS in a number of different models, both box models and global models, with different chemistry schemes and also for different planets.
**Specific Comments**

- page 1 line 16: "now include"
- page 1 line 16: "several hundred reactions"
- page 2 line 30: I'm confused by "It is also desirable to let to the user a minimum of free parameters to tune". Do you mean "desirable to give the user"?
- page 4 line 18: I think you mean "Sandu and Sander (2006)", although I can't find this reference in the reference list. I assume it is Sandu, A. and Sander, R.: Technical note: Simulating chemical systems in Fortran90 and Matlab with the Kinetic PreProcessor KPP-2.1, Atmos. Chem. Phys., 6, 187-195, doi:10.5194/acp-6-187-2006, 2006. I haven't gone through and double-checked all the other references, but I suggest that the authors do so.
- page 8 line 9: Could the authors make it clear here that when saying "To validate and evaluate the performances of ASIS and the associated numerical codes several case studies have been used. All these cases are based on the RACMOBUS chemical scheme used within the MOCAGE CTM" the authors refer to section 3 (& 4) only, as a different model and scheme are used for section 5.
- page 14 Figure 3: The quality of this figure is not very good, and also the final "s" in species is truncated on both figures.
- page 15 Figure 4: "Number of timesteps"
- page 19 line 3: I think you mean "Figure 8" here.
- page 20 Figure 8: There is a bit of an issue in terms of the figure quality, some of the bottoms of the numbers of the colour-bars are cut off at the bottom.
**• page 20 line 6: "a fixed timestep"**

- page 22 line 1: Do you mean "4.7 times"? Is this a mean over the whole 3-month run?
- page 26 Figure 11: When is this from is it a snapshot from the start/middle/end of the 150-day run?
- page 27 Figure 12: Similarly for Figure 11 when is this from with respect to the start of the run. Is it the same as for Figure 11?
- page 28 line 3: Currently the sentence implies that ASIS has been implemented in two 3D models of the Earth and two 3D models of Mars, rather than one 3D model for each.

---

## Author Comment (AC1) · 13 Feb 2017

The authors thank the referee for his comments. Since the other referees share many questions raised, a global response can be found in a supplement file.

---

## Author Comment (AC2) · 13 Feb 2017

The authors thank the referee for his comments. Since the other referees share many questions raised, a global response can be found in a supplement file.

---

## Author Comment (AC3) · 13 Feb 2017

**ASIS v1.0: an adaptative solver for the simulation of atmospheric chemistry**

Daniel Cariolle[1,2], Philippe Moinat[1], Hubert Teyssèdre[3,†], Luc Giraud[4], Béatrice Josse[3], and Franck Lefèvre[5]

[1]Climat, Environnement, Couplages et Incertitudes, UMR5318 CNRS/Cerfacs, Toulouse, France
[2]Météo-France, Toulouse, France
[3]Centre National de Recherches Météorologiques, UMR3589 CNRS/Météo-France, Toulouse, France
[4]Institut National de Recherche en Informatique et en Automatique, Talence, France
[5]Laboratoire Atmosphères, Milieux, Observations Spatiales, CNRS/UPMC/UVSQ, Paris, France
[†]deceased, April 2013

The authors would like to thank the referees for their comments and suggestions on this manuscript. We detail below the responses to the comments and the modifications that we intend to introduce in the revised manuscript.

**1. Response to general comments**

**1a. Accuracy and efficiency of the ASIS solver compared to Rosenbrock's.**

*Referee 1. First, from the paper it is not so clear why a new solver is needed. In fact, in the paper there are comments which may make the CTM modeller decide to stick to solvers like Rosenbrock. In particular I would like to see a more detailed comparison against Rosenbrock concerning run time and accuracy, for small and large numbers of chemical species.*

*Referee 2. It isn't clear to me though, why such an expensive solver should be preferred over similarly expensive solvers available through open source packages such as KPP which can provide comparable accuracy and runtimes.*

*Referee 3. I would also like more discussion of computational cost - the timestep is discussed in detail, but is rarely then compared to the overall run-time. It is certainly interesting to see where the timestep changes, but in terms of usability it would be handy to know exactly how much more time it took.*

In section 3 we have discussed in detail the accuracy of the ASIS solver compared to Rosenbrock's and Gear's type algorithms. We show that for a given relative tolerance value, Rtol, ASIS has comparable accuracy to those schemes. For low values of Rtol ASIS needs shorter timesteps to maintain comparable accuracy. For the values of Rtol that we believe small enough for atmospheric model simulations (in the range 1 to 3 %), the timesteps of ASIS and those of higher order schemes are comparable, but since ASIS requires less computation the CPU time should be comparable or lower.

We have investigated further this point and we report below several tests performed within the Matlab environment.

The following table gives the mean timesteps and CPU time for different box model configurations (the FLUX case of section 3) performed using ASIS and the ode23s code.

|          | Rtol  | Mean timestep | CPU  |
|----------|-------|---------------|------|
| ASIS     | 0,001 | 4,4           | 25,5 |
| ASIS     | 0,01  | 23,3          | 6,4  |
| ASIS     | 0,025 | 49,9          | 4,8  |
| Ode23s   | 0,001 | 39            | 58,9 |
| Ode23s   | 0,01  | 44            | 50,1 |
| Ode23s   | 0,025 | 46            | 49,9 |
| Ode23s+J | 0,001 | 39            | 6,6  |
| Ode23s+J | 0,01  | 44            | 6,2  |
| Ode23s+J | 0,025 | 46            | 6,1  |

If ode23s is used without providing a subroutine for the computation of the Jacobian of the chemical system the ode23s code is much slower than ASIS, by a factor 2 to 10. This is because the ode23s code computes by differentiation an approximation of the Jacobian. It requires more iterations with the subroutine that computes the chemical tendencies and the CPU cost is rather high.

If the routine that computes the Jacobian is provided to ode23s, the CPU cost decreases significantly (lines Ode23+J of the table) and becomes comparable to the CPU used by ASIS. At low tolerance ode23s+J is faster than ASIS, at higher tolerance the costs of ASIS and ode23s+J are comparable.

The important point to mention is that within the Matlab environment the CPU cost does not come from the linear algebra parts of the algorithms but from the evaluation of tendencies and Jacobian matrices. Therefore it

is very dependant upon the chemical system and the details of the programing of the associated subroutines.

The situation is quite different within the Fortran environment. With the Fortran version of ASIS the CPU cost for the calculation of the Jacobian (the matrix M of eq. 7) is negligible compared to the linear algebra computations. This is because the compiler handles efficiently the associated subroutine (fill_matrix) that contains frequent indirect addressing. It is not possible to evaluate if this is also the case with all the codes based on Rosenbrock's algorithm, but if it is so ASIS should perform well when the mean time steps are comparable since it needs less linear algebra computations.

In conclusion we cannot give a general statement on the computational cost of ASIS compared to Rosenbrock's solvers. It is too dependant on the computational environment, on the details of the coding of the tendencies and the Jacobian matrices associated with the chemical scheme, and on the chemical scheme itself in particular the number of species and its stiffness.

Our objective is to offer an alternative to existing solvers having in mind that ASIS should be rather effective at the rather high tolerance error that can be used by most atmospheric models. Its formulation is not complex so it can be easily coded within the environment of existing models with the help of the example available on line (see the following comment 1.d). Our approach is to avoid the use of external pre-processors that are often judged not user-friendly by modellers because the generated code has to be adapted to the chemical models (see discussion in section 7).

It is clear that if a modeller uses already an implementation of a Rosenbrock solver like KPP, the effort to turn to ASIS might be too high compared to the expected benefit. But many models do not use solvers based on Rosenbrock's or Gear's methods and we believe that ASIS could be an interesting and simple alternative for them.

In the revised manuscript we will give some indications of the CPU time to run the ASIS code and discuss the difficulty to evaluate a priori the relative efficiency of the solvers.

1b. References of other solvers

*Referee 1. There are no references given in the introduction to the general literature on chemical or differential equation solvers. An introduction should sketch the starting point of the work - the state-of-the-art - and in this way clarify how the new developments described in the paper advance this present knowledge and models. I suggest that the*

*authors add a section with references discussing the current status concerning solvers in relation to chemistry models. Several references are provided later on in the paper, but the current list is not very extended and could be expanded somewhat.*

The references to existing algorithms, solvers, and their use by chemical models are given in section 2 in connection to the discussion of the numerical treatment of the species tendency equations. By doing so we believe that the reader better sees which class of solver is associated with a given treatment of the equations. In the manuscript we give reference to the most widely used explicit (CHEMEQ, TOWSTEP) and implicit (SIS, QSSA, Rosenbrok's and Gear's) schemes.
Our objective is not to make an exhaustive review article on numerical methods and solvers but to illustrate the specificity of our scheme compared to existing solvers.

In the revised manuscript we will briefly review in the introduction the main class of solvers, and we will give in section 2 additional references on chemical models and their associated solvers.

**1c. Duration of the numerical simulations**

*Referee 2. I find it a shame that only 3 month simulations are performed with the global models using the ASIS scheme. It would be very interesting to know what effect the replacement of the chemical solver would have on many other features of global atmospheric chemistry, such as the global oxidising capacity and related aspects such as the methane lifetime and tropospheric ozone budget.*

*Referee 3. The run length is also a bit short for most cases. The 1 day simulations for the box model are very short, especially when it appears that the A2 case is diverging from the R1/G1/A1 cases. Have these simulations been run for longer, and if so, how do the results of A2 (& A3) evolve? Also, A3 is not plotted at all, but these settings are then used for the MOCAGE simulations. The A3 results should also be included in the plots in Figures 1 & 2 (or plotted separately with a larger scale if required).*
*The GCM lengths are slightly better (3 months and 150 days), but still short. Are there plans to do longer runs with a full evaluation and budget analysis? The results presented here have highlighted deficiencies in the existing models, but a full analysis on longer simulations would be required to properly validate ASIS, as opposed to this paper which describes its implementation.*

The objective of the reported simulations is to present the characteristics of the ASIS solver in terms of accuracy and adaptability to various chemical schemes and situations.

The 1-day simulation with the box model is long enough to evaluate the accuracy of the solver. We have extended the simulations up to 3 days for the FLUX case and the results obtained are fully consistent with the 1-day simulation. The next figures show for example results of the time evolution of $O_3$ and $NO_2$ concentrations in a 3-day extension of the A3 experiment (with the largest tolerance, 0.025) and its relative difference with an extended G1 experiment.

[Figure]

As can be seen there is no specific trend in relative differences between the species, the differences remain in the range of the chosen relative tolerance. In the revised manuscript we will include in figure 1 the results of experiment A3.

The 3D simulations illustrate the benefit of using a solver like ASIS that has a controlled accuracy and is mass conserving. The 3-month simulation with MOCAGE is short, but long enough to point out the benefits from the ASIS use. We agree that a more fully validation of

MOCAGE+ASIS would require longer simulations, in particular to assess the impact of ASIS on the longer-lived species. This is however beyond the scope of this article. Multiyear simulations of MOCAGE+ASIS are planed in the near future along with simulation of the C-IFS model (Flemming et al., 2015) with the RACMOBUS chemical scheme.

1d Code availability

*Referee 1. - p29, code availability: "The ASIS code is property of the CERFACS and includes libraries that belong to other holders." Does this imply any restrictions if other parties want to use the code? Is there a kind of license for using the code?*

*Referee 2. if the authors do not seriously intend to make their code generally available to the community, and are content with limiting its use to in-house applications, differentiating their product from other alternatives in this way is arguably outside the scope of the manuscript.*

After discussions with the holders of the different parts of the code it was agreed that the 0D Fortran code used in section 3 will be made generally available on the CERFACS's server.

1e . Synthesis of the simulations

*Referee 3. Due to the large number of tests performed it was a little difficult to keep track of the settings used in each case. I would recommend giving a master table (or tables) giving the configuration for each shorthand used in the plots (e.g. R1, G1, A1, A2, A3, A4, MR, MA, EB, ASIS etc.) and what the settings are used for each (e.g. values for ATOL, RTOL, using ode23s, ode15s, DGESV, GS, GMRES etc.), the experiment run (e.g. FLUX, STRATO, MOCAGE, Mars Box Model, LMD Mars GCM etc.), and also the chemistry scheme used (e.g. RACMOBUS, Mars). This would be especially helpful for comparing between sections, as it can be difficult to pick out this mass of information from the text.*

We will extend table 4 to give the information required by Referee 3.

2. Response to specific comments

All the typos and english shortcommings will be addressed in the revised manuscript and are not detailed hereafter.

Equally, we will improve the quality of the figures as suggested by referee 3.

*Referee 3. p2 line 30: I'm confused by "It is also desirable to let to the user a minimum of free parameters to tune". Do you mean "desirable to give the user"?*

Yes, the idea is to give to the user a choice in a limited number of parameters that control the accuracy of the solution.

*Referee 1.  p3, line 25: The second term on the left is diagonal. Please explain why ? Diagonal in which space?*

$L(t, C)$ being completely explicit, the matrix $(I + L(t, C) \delta t)$ is diagonal by construction.

*Referee 1. p3, line 27: "mass conservation is not maintained". Can you add a reference or text book? Is it possible to describe in one line how the reader may understand that there is no conservation of mass?*

The mass conservation is not maintained when the species tendencies associated to a given reaction are different after time discretisation. This is for instance the case with the simple BDF scheme. We will recall this in the revised manuscript.

*Referee 3. p4 line 18: I think you mean "Sandu and Sander (2006)", although I can't find this reference in the reference list. I assume it is Sandu, A. and Sander, R.: Technical note: Simulating chemical systems in Fortran90 and Matlab with the Kinetic PreProcessor KPP-2.1, Atmos. Chem. Phys., 6, 187-195, doi:10.5194/acp-6-187-2006, 2006. I haven't gone through and double-checked all the other references, but I suggest that the authors do so.*

Yes this is the right reference. We will double-check again all the other references.

*Referee 1. p5, line 27: Why is beta >= 1 required? Should this be beta > 0?*
Beta > 0 is enough from a mathematical point of view, but to better discriminate between implicit and explicit parts for the species tendencies beta >1 is more appropriate. We have tested values for beta >1 but not investigated 0 < beta < 1.

*Referee 1.- Fig.6. The colour scale is unclear: Does the colour between 1 and 2 mean there are two substeps or one substep? I assume "number of sub-timesteps" can only take integer values.*

The first colour (dark blue) corresponds to 1 sub-timestep. The coulour scale intervals should be read ] lower value, higher value ].
The number of sub-timesteps can of course only take integer values. However we show interpolated values from model levels (which are function of ground pressure) to pressure levels (50hPa and 540hPa) and interpolated values are generally not integers.

*Referee 3. p22 line 1: Do you mean "4.7 times"? Is this a mean over the whole 3-month run?*
Yes it is 4.7 times, calculated over the 3-month run.

*Referee 1. p25, line 32: "4 times smaller". How can I see this from the figure 12, which has a scale ranging between -30 to 30%?*

In order to highlight differences obtained during the day, the colour scale of Figure 12 is limited to maximum and minimum values of ±30%. This is now mentioned in the legend. The related text in the body of the manuscript will be modified as follows:
"Regarding OH, the GCM results confirm the poor description of the HOx chemistry by the EB scheme at the terminators. At night, OH values calculated by EB are more than 30% smaller than with ASIS".

*Referee 3.*
*p26 Figure 11: When is this from - is it a snapshot from the start/middle/end of the 150-day run?*
It is a snapshot at the end of the 150-day run, now indicated in the revised legend of Figure 11.

*page 27 Figure 12: Similarly for Figure 11 - when is this from with respect to the start of the run. Is it the same as for Figure 11?*

Yes similarly to Figure 11 this is a snapshot at the end of the 150-day run. This will be indicated in the revised legend of Figure 12.

*Referee 1. p28: I was wondering if ASIS could be used for adjoint (4D-Var) type of applications?*

In theory, the adjoint of the ASIS code can be developed. It requires that all the intermediate calculations  (sub-timesteps, matrix M evaluations,

...) be stored. Then the adjoint of the successive linear operators can be derived if a direct method (for instance based on a LU decomposition) is used to solve the linear systems (eq. 7). The situation is more complex if an iterative solver is used.

However we do not plan to develop the adjoint of the ASIS code, we are alternatively exploring ensemble methods (Emili et al., 2016) for assimilation applications.

References

Emili, E., Gürol, S., and Cariolle, D.: Accounting for model error in air quality forecasts: an application of 4DEnVar to the assimilation of atmospheric composition using QG-Chem 1.0, Geosci. Model Dev., 9, 3933-3959, doi:10.5194/gmd-9-3933-2016, 2016.

Flemming, J., Huijnen, V., Arteta, J., Bechtold, P., Beljaars, A., Blechschmidt, A.-M., Diamantakis, M., Engelen, R. J., Gaudel, A., Inness, A., Jones, L., Josse, B., Katragkou, E., Marecal, V., Peuch, V.-H., Richter, A., Schultz, M. G., Stein, O., and Tsikerdekis, A.: Tropospheric chemistry in the Integrated Forecasting System of ECMWF, Geosci. Model Dev., 8, 975-1003, doi:10.5194/gmd-8-975-2015, 2015.

---

## Author Response (AR1)

**ASIS v1.0: an adaptive solver for the simulation of atmospheric chemistry**

Daniel Cariolle1,2, Philippe Moinat1, Hubert Teyssèdre3,†, Luc Giraud4, Béatrice Josse3, and Franck Lefèvre5 1Climat, Environnement, Couplages et Incertitudes, UMR5318 CNRS/Cerfacs, Toulouse, France 2Météo-France, Toulouse, France 3Centre National de Recherches Météorologiques, UMR3589 CNRS/Météo-France, Toulouse, France 4Institut National de Recherche en Informatique et en Automatique, Talence, France 5Laboratoire Atmosphères, Milieux, Observations Spatiales, CNRS/UPMC/UVSQ, Paris, France †deceased, April 2013

Dear Editor,

We hereby submit a revised version of the manuscript gmd-2016-281 entitled « ASIS v1.0: an adaptive solver for the simulation of atmospheric chemistry » for consideration of publication in GMD. We appreciate the careful and insightful reviews from the anonymous referees. We recall here the response to the comments made by the referees and detail the modifications that we have made in the manuscript. Follows the new version of the article in difference mode that clearly shows the modifications introduced in the revised article.

Best regards,

D. Cariolle

**1. Response to general comments**

**Comment 1a. Accuracy and efficiency of the ASIS solver compared to Rosenbrock's.**

Referee 1. First, from the paper it is not so clear why a new solver is needed. In fact, in the paper there are comments which may make the CTM modeller decide to stick to solvers like Rosenbrock. In particular I would like to see a more detailed comparison against Rosenbrock concerning run time and accuracy, for small and large numbers of chemical species.

Referee 2. It isn't clear to me though, why such an expensive solver should be preferred over similarly expensive solvers available through

open source packages such as KPP which can provide comparable accuracy and runtimes.

Referee 3. I would also like more discussion of computational cost - the timestep is discussed in detail, but is rarely then compared to the overall run-time. It is certainly interesting to see where the timestep changes, but in terms of usability it would be handy to know exactly how much more time it took.

**Response**

In section 3 we have discussed in detail the accuracy of the ASIS solver compared to Rosenbrock's and Gear's type algorithms. We show that for a given relative tolerance value, Rtol, ASIS has comparable accuracy to those schemes. For low values of Rtol ASIS needs shorter timesteps to maintain comparable accuracy. For the values of Rtol that we believe small enough for atmospheric model simulations (in the range 1 to 3 %), the timesteps of ASIS and those of higher order schemes are comparable, but since ASIS requires less computation the CPU time should be comparable or lower.

We have investigated further this point and we report below several tests performed within the Matlab environment.

The following table gives the mean timesteps and CPU time for different box model configurations (the FLUX case of section 3) performed using ASIS and the ode23s code.

|          | Rtol  | Mean
timestep | CPU  |
|----------|-------|------------------|------|
| ASIS     | 0,001 | 4,4              | 25,5 |
| ASIS     | 0,01  | 23,3             | 6,4  |
| ASIS     | 0,025 | 49,9             | 4,8  |
| Ode23s   | 0,001 | 39               | 58,9 |
| Ode23s   | 0,01  | 44               | 50,1 |
| Ode23s   | 0,025 | 46               | 49,9 |
| Ode23s+J | 0,001 | 39               | 6,6  |
| Ode23s+J | 0,01  | 44               | 6,2  |
| Ode23s+J | 0,025 | 46               | 6,1  |

If ode23s is used without providing a subroutine for the computation of the Jacobian of the chemical system the ode23s code is much slower than ASIS, by a factor 2 to 10. This is because the ode23s code computes by differentiation an approximation of the Jacobian. It requires more iterations with the subroutine that computes the chemical tendencies and the CPU cost is rather high.

If the routine that computes the Jacobian is provided to ode23s, the CPU cost decreases significantly (lines Ode23+J of the table) and becomes comparable to the CPU used by ASIS. At low tolerance ode23s+J is faster than ASIS, at higher tolerance the costs of ASIS and ode23s+J are comparable.

The important point to mention is that within the Matlab environment the CPU cost does not come from the linear algebra parts of the algorithms but from the evaluation of tendencies and Jacobian matrices. Therefore it is very dependent upon the chemical system and the details of the programing of the associated subroutines.

The situation is quite different within the Fortran environment. With the Fortran version of ASIS the CPU cost for the calculation of the Jacobian (the matrix M of eq. 7) is negligible compared to the linear algebra computations. This is because the compiler handles efficiently the associated subroutine (fill\_matrix) that contains frequent indirect addressing. It is not possible to evaluate if this is also the case with all the codes based on Rosenbrock's algorithm, but if it is so ASIS should perform well when the mean time steps are comparable since it needs less linear algebra computations.

In conclusion we cannot give a general statement on the computational cost of ASIS compared to Rosenbrock's solvers. It is too dependant on the computational environment, on the details of the coding of the tendencies and the Jacobian matrices associated with the chemical scheme, and on the chemical scheme itself in particular the number of species and its stiffness.

Our objective is to offer an alternative to existing solvers having in mind that ASIS should be rather effective at the rather high tolerance error that can be used by most atmospheric models. Its formulation is not complex so it can be easily coded within the environment of existing models with the help of the example available on line (see the following comment 1.d). Our approach is to avoid the use of external pre-processors that are often judged not user-friendly by modellers because the generated code has to be adapted to the chemical models (see discussion in section 7). It is clear that if a modeller uses already an implementation of a Rosenbrock solver like KPP, the effort to turn to ASIS might be too high compared to the expected benefit. But many models do not use solvers based on Rosenbrock's or Gear's methods and we believe that ASIS could be an interesting and simple alternative for them.

**Change in the manuscript:**

In section 3.4 we have extended the table 4 to introduce the results of the simulations discussed above. In particular the ratio of CPU relative to the R1 simulation is given.

In addition we have introduced in the same section (page 11 and 12, from lines 284 to 297) a detailed discussion of the performances of ASIS relative to the Rosenbrock ode23s code.

**Comment 1b. References of other solvers**

Referee 1. There are no references given in the introduction to the general literature on chemical or differential equation solvers. An introduction should sketch the starting point of the work - the state-of-theart - and in this way clarify how the new developments described in the paper advance this present knowledge and models. I suggest that the authors add a section with references discussing the current status concerning solvers in relation to chemistry models. Several references are provided later on in the paper, but the current list is not very extended and could be expanded somewhat.

**Response:**

The references to existing algorithms, solvers, and their use by chemical models are given in section 2 in connection to the discussion of the numerical treatment of the species tendency equations. By doing so we believe that the reader better sees which class of solver is associated with a given treatment of the equations. In the manuscript we give reference to the most widely used explicit (CHEMEQ, TOWSTEP, QSSA) and implicit (SIS, Rosenbrok's and Gear's) schemes.

Our objective is not to make an exhaustive review article on numerical methods and solvers but to illustrate the specificity of our scheme compared to existing solvers.

**Change in the manuscript:**

In the introduction (page 2, from lines 34 to 40) we have introduced references to the main class of solvers, implicit versus explicit, and multi-step versus one-step algorithms.

In section 2.1 additional references to chemical models and their associated solvers have been added (page 4, lines 99, 103, 104).

**Comment 1c. Duration of the numerical simulations**

Referee 2. I find it a shame that only 3 month simulations are performed with the global models using the ASIS scheme. It would be very interesting to know what effect the replacement of the chemical solver would have on many other features of global atmospheric chemistry, such as the global oxidising capacity and related aspects such as the methane lifetime and tropospheric ozone budget.

Referee 3. The run length is also a bit short for most cases. The 1 day simulations for the box model are very short, especially when it appears that the A2 case is diverging from the R1/G1/A1 cases. Have these simulations been run for longer, and if so, how do the results of A2 (& A3) evolve? Also, A3 is not plotted at all, but these settings are then used for the MOCAGE simulations. The A3 results should also be included in the plots in Figures 1 & 2 (or plotted separately with a larger scale if required).

The GCM lengths are slightly better (3 months and 150 days), but still short. Are there plans to do longer runs with a full evaluation and budget analysis? The results presented here have highlighted deficiencies in the existing models, but a full analysis on longer simulations would be required to properly validate ASIS, as opposed to this paper which describes its implementation.

**Response:**

The objective of the reported simulations is to present the characteristics of the ASIS solver in terms of accuracy and adaptability to various chemical schemes and situations.

The 1-day simulation with the box model is long enough to evaluate the accuracy of the solver. We have extended the simulations up to 3 days for the FLUX case and the results obtained are fully consistent with the 1-day simulation. The next figures show for example results of the time evolution of  $O_3$  and  $NO_2$  concentrations in a 3-day extension of the A3 experiment (with the largest tolerance, 0.025) and its relative difference with an extended G1 experiment.

As can be seen there is no specific trend in relative differences between the species, the differences remain in the range of the chosen relative tolerance. In the revised manuscript we will include in figure 1 the results of experiment A3.

The 3D simulations illustrate the benefit of using a solver like ASIS that has a controlled accuracy and is mass conserving. The 3-month simulation with MOCAGE is short, but long enough to point out the benefits from the ASIS use. We agree that a more fully validation of MOCAGE+ASIS would require longer simulations, in particular to assess the impact of ASIS on the longer-lived species. This is however beyond the scope of this article. Multiyear simulations of MOCAGE+ASIS are planed in the near future along with simulation of the C-IFS model (Flemming et al., 2015) with the RACMOBUS chemical scheme.

**Change in the manuscript:**

We have included the results of experiment A3 in the figures 1 and 2.

**Comment 1d. Code availability**

Referee 1. - p29, code availability: "The ASIS code is property of the CERFACS and includes libraries that belong to other holders." Does this imply any restrictions if other parties want to use the code? Is there a kind of license for using the code?

Referee 2. if the authors do not seriously intend to make their code generally available to the community, and are content with limiting its use to in-house applications, differentiating their product from other alternatives in this way is arguably outside the scope of the manuscript.

**Response:**

After discussions with the holders of the different parts of the code it was agreed that the 0D Fortran code used in section 3 will be made generally available on the CERFACS's server.

**Change in the manuscript:**

In section 7 (page 30, lines 569 to 570) we give indication on how to obtain a copy of the Fortran code: as a supplementary file to the article or on the CERFACS server.

**Comment 1e . Synthesis of the simulations**

Referee 3. Due to the large number of tests performed it was a little difficult to keep track of the settings used in each case. I would recommend giving a master table (or tables) giving the configuration for each shorthand used in the plots (e.g. R1, G1, A1, A2, A3, A4, MR, MA, EB, ASIS etc.) and what the settings are used for each (e.g. values for ATOL, RTOL, using ode23s, ode15s, DGESV, GS, GMRES etc.), the experiment run (e.g. FLUX, STRATO, MOCAGE, Mars Box Model, LMD Mars GCM etc.), and also the chemistry scheme used (e.g. RACMOBUS, Mars). This would be especially helpful for comparing between sections, as it can be difficult to pick out this mass of information from the text.

**Response:**

We will extend table 4 to give the information required by Referee 3.

**Change in the manuscript:**

The table 4 has been extended to include a summary of the input/output of the simulations for the FLUX case.

For the STRATO case the name of the simulation was change to mirror the name used for the FLUX cases: AS2 of the STRATO refers to the A2 simulation of the FLUX case using the same settings for ASIS.

**2. Response to specific comments**

The typos and english shortcommings have been addressed in the revised manuscript and are not detailed hereafter.

Equally, we will improve the quality of the figures as suggested by referee 3.

**Change in the manuscript:**

Some figures have been redrawn to increase their quality. One figure was deleted (figure 3 of the original manuscript) in order to shorten the

article and because its content was not essential for the understanding of the model results.

**Comment by Referee 3**. *p2 line 30: I'm confused by "It is also desirable to let to the user a minimum of free parameters to tune". Do you mean "desirable to give the user"?*

**Response:**

Yes, the idea is to give to the user a choice in a limited number of parameters that control the accuracy of the solution.

**Change in the manuscript:**

The sentence has been changed (page 3 line 59).

**Comment by Referee 1**. p3, line 25: The second term on the left is diagonal. Please explain why ? Diagonal in which space?

**Response:**

L(t, C) being completely explicit, the matrix  $(I + L(t, C) \delta t)$  is diagonal by construction.

**Change in the manuscript:**

The sentence has been changed (page 4 line 90).

Comment by Referee 1. p3, line 27: "mass conservation is not maintained". Can you add a reference or text book? Is it possible to describe in one line how the reader may understand that there is no conservation of mass?

**Response:**

The mass conservation is not maintained when the species tendencies associated to a given reaction are different after time discretisation. This is for instance the case with the simple BDF scheme. We will recall this in the revised manuscript.

**Change in the manuscript:**

The sentence "The mass conservation is however not maintained due to the fact that for a given reaction between two species the value of the associated tendency is different for each species." is added page 4 line 92.

**Comment by Referee 3**. p4 line 18: I think you mean "Sandu and Sander (2006)", although I can't find this reference in the reference list. I assume it is Sandu, A. and Sander, R.: Technical note: Simulating chemical systems in Fortran90 and Matlab with the Kinetic PreProcessor KPP-2.1, Atmos. Chem. Phys., 6, 187-195, doi:10.5194/acp-6-187-2006, 2006. I haven't gone through and double-checked all the other references, but I suggest that the authors do so. **Response:**

Yes this is the right reference. We will double-check again all the other references.

**Change in the manuscript:**

The reference has been added (page 33 line 673).

**Comment by Referee 1**. p5, line 27: Why is beta >= 1 required? Should this be beta > 0?

**Response:**

Beta > 0 is enough from a mathematical point of view, but to better discriminate between implicit and explicit parts for the species tendencies beta >1 is more appropriate. We have tested values for beta >1 but not investigated 0 < beta < 1.

**Change in the manuscript:**

This point is briefly discussed in the revised manuscript (page 6, line 164)

**Comment by Referee 1**.- Fig.6. The colour scale is unclear: Does the colour between 1 and 2 mean there are two substeps or one substep? I assume "number of sub-timesteps" can only take integer values.

**Response:**

The first colour (dark blue) corresponds to 1 sub-timestep. The colour scale intervals should be read ] lower value, higher value ].

The number of sub-timesteps can of course only take integer values. However we show interpolated values from model levels (which are function of ground pressure) to pressure levels (50hPa and 540hPa) and interpolated values are generally not integers.

**Comment by Referee 3**. p22 line 1: Do you mean "4.7 times"? Is this a mean over the whole 3-month run?

**Response:**

Yes it is 4.7 times, calculated over the 3-month run.

**Change in the manuscript:**

It is now specified "4.7 times" in the revised manuscript (page 22 line 427).

**Comment by Referee 1**. p25, line 32: "4 times smaller". How can I see this from the figure 12, which has a scale ranging between -30 to 30%? **Response:**

In order to highlight differences obtained during the day, the colour scale of Figure 12 is limited to maximum and minimum values of  $\pm 30\%$ . This is now mentioned in the legend. The related text in the body of the manuscript will be modified as follows:

"Regarding OH, the GCM results confirm the poor description of the HOx chemistry by the EB scheme at the terminators. At night, OH values calculated by EB are more than 30% smaller than with ASIS".

**Change in the manuscript:**

The sentence has been rephrased "At night, OH values calculated by EB are more than 30% smaller than with ASIS " (page 27, line 521).

**Comment by Referee 3**

p26 Figure 11: When is this from - is it a snapshot from the start/middle/end of the 150-day run?

**Response:**

It is a snapshot at the end of the 150-day run, now indicated in the revised legend of Figure 11.

**Change in the manuscript:**

The legend has been changed for what is now figure 10.

**Comment by Referee 3**

page 27 Figure 12: Similarly for Figure 11 - when is this from with respect to the start of the run. Is it the same as for Figure 11?

**Response:**

Yes similarly to Figure 11 this is a snapshot at the end of the 150-day run. This will be indicated in the revised legend of Figure 12.

**Change in the manuscript:**

The legend has been changed for what is now figure 11.

**Comment by Referee 1**. p28: I was wondering if ASIS could be used for adjoint (4D-Var) type of applications?**

**Response:**

In theory, the adjoint of the ASIS code can be developed. It requires that all the intermediate calculations (sub-timesteps, matrix M evaluations,

...) be stored. Then the adjoint of the successive linear operators can be derived if a direct method (for instance based on a LU decomposition) is used to solve the linear systems (eq. 7). The situation is more complex if an iterative solver is used.

However we do not plan to develop the adjoint of the ASIS code, we are alternatively exploring ensemble methods (Emili et al., 2016) for assimilation applications.

**Change in the manuscript:**

No change has been made; this subject is out of the scope of the article (but of interest for the referee).

[revised manuscript text omitted]
(^{1}D), O(^{3}P), O_{2}, O_{3},$ N, N2O, NO, NO2, NO3, N2O5, HNO2, HNO4, HNO3(gas&solid), CH4, CH2O, CH3, CH4O, CH3O, CHO, CH4O2, CH3O2, CO, CO2  $H_2, H_2O(gas\&solid), H, OH, HO_2, H_2O_2,$ SO2, H2SO4, DMS, SULFATE CCl4, CFC-(11&12&113&114&115), HCFC-22, HA-(1202&1211&1301), CH3Cl, CHCl3, CH3CCl3, Cl, Cl2, ClO, OClO, ClO2, Cl2O2, HOCl, HCl, ClONO2 CH3Br, CHBr3, Br, Br2, BrO, HBr, HOBr, BrONO2, BrCl ACO3, ADDC, ADDT, ADDX, ALD, API, APIP, CLS, CSLP, DCB, DIEN, ETE, ETEP, ETH, ETHP, GLY, HC3, HC3P, HC5, HC5P, HC8, HC8P, HKET, ISO, ISOP KET, KETP, LIM, LIMP, MACR, MGLY, MO2 OLI, OLIP, OLND, OLNN, OLT, OLTP, ONIT, OP1, OP2 PAA, PAN, PHO, TCO3, TOL, TOLP, TPAN, UDD, XO2, XYL, XYLP

Two test cases are used to evaluate the accuracy and performance of the ASIS scheme. The first one is based on the FLUX test case described by Crassier et al. (2000). It corresponds to a ground level situation in a an urban polluted area. The list of species and fluxes emitted at the surface is given in table 2. The emissions are injected in a boundary layer with a 2000 m constant thickness weighted by an emission factor of 0.6. This leads to a constant tendency F in Eq. (4) for the emitted species. The initial concentrations are given in table 3, the atmospheric temperature is set to 298 K and the ground pressure is 1000 hPa.

250 The second case, STRATO, is representative of situations encountered in the middle stratosphere. The initial concentrations for this case are given in table 3. The atmospheric temperature is 215 K and the pressure is 50 hPa. For both cases the integration starts at midnight, stops 24 h after, and the photodissociation rates are updated every 15 minutes.

**3.1 The FLUX case**

To assess the performances of ASIS two reference simulations have been obtained for the FLUX case using Rosenbrock's and 255 Gear's BDF solvers (referred hereafter as R1 and G1). Those solvers use respectively the ode23s and ode15s codes from the Matlab ODE suite (Shampine and Reichelt, 1997, Ashino et al., 2000). For the Rosenbrock's scheme a 3 stage algorithm is used and the simulations are third order accurate. For the Gear's scheme the third order accurate option was also chosen. The

| Species                     | Emission                                     |
|-----------------------------|----------------------------------------------|
|                             | $(10^{10} \text{ molecules } cm^{-2}s^{-1})$ |
| NO                          | 121.29                                       |
| СО                          | 2500                                         |
| $\mathrm{CH}_4$             | 802                                          |
| ETH                         | 6.25                                         |
| HC3                         | 37.67                                        |
| HC5                         | 44.43                                        |
| HC8                         | 19.14                                        |
| ETE                         | 22.33                                        |
| OLT                         | 39.67                                        |
| OLI                         | 6.37                                         |
| TOL                         | 9.02                                         |
| $\mathrm{CH}_{2}\mathrm{O}$ | 5.77                                         |
| ALD                         | 14.45                                        |
| KET                         | 5.70                                         |
| XYL                         | 14.55                                        |
| $\operatorname{CSL}$        | 3.68                                         |

Table 2. VOC emissions in the FLUX test case

Table 3. Initial conditions for the FLUX and STRATO test cases

| Species            | STRATO          | FLUX             |  |
|--------------------|-----------------|------------------|--|
|                    | vmr             | vmr              |  |
| $O_3$              | $1.0 \ 10^{-6}$ | $50 \ 10^{-9}$   |  |
| $\mathrm{CO}_2$    | $330 \ 10^{-6}$ | $330 \ 10^{-6}$  |  |
| $N_2O$             | $300 \ 10^{-9}$ | $310 \ 10^{-9}$  |  |
| NO                 | $1.0 \ 10^{-9}$ | $2.0 \; 10^{-9}$ |  |
| $NO_2$             | $0.310^{-6}$    | $1.0 \; 10^{-9}$ |  |
| $\mathrm{HNO}_3$   | $4.0\ 10^{-9}$  | $0.5 \ 10^{-9}$  |  |
| $\mathrm{CH}_4$    | $1.4 \ 10^{-6}$ | $1.6 \ 10^{-6}$  |  |
| СО                 | $20 \; 10^{-9}$ | $150 \; 10^{-9}$ |  |
| HCl                | $2.5 \ 10^{-9}$ | $1.0 \ 10^{-12}$ |  |
| $\mathrm{ClONO}_2$ | $0.3 \ 10^{-9}$ |                  |  |
| BrO                | $15 \ 10^{-12}$ | $1.0 \ 10^{-13}$ |  |

In the most accurate simulations R1 and G1 (see table 4) the relative tolerance RTOL was is set to 0.001 and the absolute

| is the CPU time used in | n each simulation relative to R1. The simulations are performed in th | e Matlab environment. |
|-------------------------|-----------------------------------------------------------------------|-----------------------|
|                         |                                                                       |                       |

**Table 4.** Mean timesteps List of the different settings used by the 0D model for the FLUX test case,  $\delta t_m$  is the mean time step and CPU-Ratio

[revised manuscript text omitted]